# Solvent Effect on the Regulation of Urea Hydrolysis Reactions by Copper Complexes

**Caio B. Castro** [1], **Rafael G. Silveira** [2], **Felippe M. Colombari** [3], **André Farias de Moura** [4], **Otaciro R. Nascimento** [5] and **Caterina G. C. Marques Netto** [1,*]

[1] Laboratório de Metaloenzimas e Biomiméticos, Departamento de Química, Universidade Federal de São Carlos (UFSCar), Rod. Washington Luiz, km 235, São Carlos, São Paulo 13565-905, Brazil; caiobezerradecastro@gmail.com

[2] Instituto Federal Goiano, Rod. GO, km 154, s/n 03, Ceres, Goiás 76300-000, Brazil; rafael.silveira@ifgoiano.edu.br

[3] Laboratório Nacional de Nanotecnologia, Rua Giuseppe Máximo Scolfaro, 10.000, Polo II de Alta Tecnologia de Campinas, Campinas, SP 13083-100, Brazil; colombarifm@hotmail.com

[4] Laboratório de Química Teórica, Departamento de Química, Universidade Federal de São Carlos (UFSCar), Rod. Washington Luiz, km 235, São Carlos, São Paulo 13565-905, Brazil; moura@ufscar.br

[5] Instituto de Física de São Carlos, Universidade de São Paulo (USP), Av. João Dagnone, 1100, Jardim Santa Angelina, São Carlos, São Paulo 13563-120, Brazil; ciro@ifsc.usp.br

[*] Correspondence: caterina@ufscar.br; Tel.: +55-16-3306-6653

**Abstract:** Abiotic allosterism is most commonly observed in hetero-bimetallic supramolecular complexes and less frequently in homo-bimetallic complexes. The use of hemilabile ligands with high synthetic complexity enables the catalytic center by the addition or removal of allosteric effectors and simplicity is unusually seen in these systems. Here we describe a simpler approach to achieve kinetic regulation by the use of dimeric Schiff base copper complexes connected by a chlorido ligand bridge. The chlorido ligand acts as a weak link between monomers, generating homo-bimetallic self-aggregating supramolecular complexes that generate monomeric species in different reaction rates depending on the solvent and on the radical moiety of the ligand. The ligand exchange was observed by electron paramagnetic resonance (EPR) and conductivity measurements, indicating that complexes with ligands bearing methoxyl ($Cu^{II}L2$) and ethoxyl ($Cu^{II}L5$) radicals were more prone to form dimeric complexes in comparison to ligands bearing hydrogen ($Cu^{II}L1$), methyl ($Cu^{II}L3$), or t-butyl ($Cu^{II}L4$) radicals. The equilibrium between dimer and monomer afforded different reactivities of the complexes in acetonitrile/water and methanol/water mixtures toward urea hydrolysis as a model reaction. It was evident that the dimeric species were inactive and that by increasing the water concentration in the reaction medium, the dimeric structures dissociated to form the active monomeric structures. This behavior was more pronounced when methanol/water mixtures were employed due to a slower displacement of the chlorido bridge in this medium than in the acetonitrile/water mixtures, enabling the reaction kinetics to be evaluated. This effect was attributed to the preferential solvation shell by the organic solvents and in essence, an upregulation behavior was observed due to the intrinsic nature of the complexes to form dimeric structures in solution that could be dismantled in the presence of water, indicating their possible use as water-sensors in organic solvents.

**Keywords:** copper complexes; chlorido ligand displacement; catalysis regulation; Schiff base ligands; urea hydrolysis; supramolecular chemistry

## 1. Introduction

Allosterism is commonly observed in proteins that suffer conformational changes induced by ligand binding to an orthosteric site, producing an activation, inhibition, or regulation of the enzymatic activity [1]. Allosteric enzymes optimize the interactions between the ligand and host to tune the populations of active and inactive states for a specific metabolic function [2]. Therefore, allosterism control leads to a high specificity of these enzymes [3,4], which can inspire the design and development of supramolecular devices with high selectivity toward a substrate or analyte. Therein, abiotic allosteric catalysts have shown that metal ions can induce the control of conformation and reactivity of dinuclear catalytic sites [5–7]. The use of redox switching [8] and anion binding affinity [9–11] has also been explored in allosteric coordination chemistry, in which the weak-link approach (WLA) is afforded by the employment of hemilabile ligands to obtain systems that are stimuli-responsive [10].

The WLA approach allows for reversibility in enzyme mimics, leading to allosteric responses [12] as well as to the development of small-molecule sensors such as enzyme-linked immunosorbent assays (ELISA) [13,14] and polymerase chain reaction (PCR) [15]. In the WLA approach, systems with ditopic ligands are commonly used since they enable the interaction of the complex with other molecular species such as anions and cations, causing the change in the overall conformation of the complex [16]. This change in conformation induces an alteration in the properties of one of the metal centers. Classical examples of systems employing WLA in an allosteric conformational manner can be found in the literature, in which the molecular structure [17], binding specificity [18], and catalytic activity [19,20] are modulated upon the interaction with a regulator.

Ligands based on Schiff bases are common moieties in complexes bearing allosteric behaviors [12,21,22] and homo-bimetallic complexes have been shown to be effective in several catalytic reactions [23–26], with special attention to the supramolecular assemblies forming dimeric structures that exhibit significant rate acceleration when compared to the corresponding monomeric catalyst [27,28]. However, these systems are based on coordination compounds with rather sophisticated structures and it would be valuable to find simpler structural complexes bearing abiotic allosterism or catalytic regulation. In this aspect, dimeric or polymeric structures can be easily achieved with chlorido bridges [29–33], which can suffer ligand substitution reactions to form monomeric species in solution, enabling an easy approach to obtain different reactivities of the complex in an allosteric manner.

In this work, a simple coordination system was designed to achieve allosteric behavior by the regulation of the equilibrium between monomeric and dimeric species. For this purpose, Schiff base ligands based on *L*-proline were designed and coordinated to Cu$^{II}$, as shown in Figure 1. The equilibrium between monomeric and dimeric species was dependent on the solvent mixture used in the reaction, enabling their use in a model reaction. As a proof-of-concept, urea hydrolysis was performed by these complexes, and water was shown to act as an allosteric regulator, since it induced the formation of active monomeric structures. Hence, we demonstrate that coordination systems with less complexity can also be used in regulatory reactions, serving as an inspiration to the development of cheaper sensors.

**Figure 1.** Synthesis of Cu$^{II}$L1-L5(X) complexes. * Note: The perchlorate anion was used only for ligand L2.

## 2. Materials and Instruments

The reagents were of analytical grade and were used without prior purification. Thionylchloride ($SOCl_{2(l)}$) used in the esterification of L-proline, as described in the literature, was previously distilled under argon. The solvents used in the synthesis and experiments were previously distilled. High Resolution Mass Spectra (HRMS) were obtained with a MICROTOF–Bruker Daltonics (Billerica, MA, USA) in the positive mode. Nebulizer: 0.3 bar, Dry gas: 4 mL/min, temperature: 180 °C, High Voltage: 4500 V. Nuclear Magnetic Resonance (NMR) spectra were recorded on a BRUKER DRX 400 MHz using $CDCl_3$ as the solvent. X-ray diffraction data were obtained on a Rigaku XtaLAB Mini diffractometer (Rigaku, Tokio, Japan) with an X-ray generator operating at 50 kV and 12 mA with a graphite monochromatic Mo-K ($\lambda = 0.71073$ Å) using the Olex2 program (version Olex2 v1.3© OlexSys Ltd. 2004–2020, Chemistry Department, Durham University, Durham, UK). EPR spectra were obtained on a Varian E109 EPR X-Band (Varian, Palo Alto, CA, USA) using the rectangular cavity field modulation at 100 kMz. Parameters: microwave power of 20 mW, modulation amplitude of 0.4 mT peak to peak, gain adjustable for each sample, field scan of 160 mT, time constant of 0.064 s, scan time of 3 min. For the measurements of liquid, $N_2$ was used in the Dewar immersion method. An EPR standard was used to calibrate the magnetic field (MgO crystal: $Cr^{III}$ g = 1.9797) and the resonance frequency was measured with a microwave frequency meter. FTIR spectra were recorded on a Bomen-Michelson FT model MB-102 spectrometer (ABB BOMEM, Quebec, QC, Canada) in the 4000–200 $cm^{-1}$ region. In situ FTIR were recorded on a Nicolet 6700 FTIR spectrometer equipped with a Mercury-Cadmium-Telluride (MCT) detector using p-polarized light and employed a 60 $CaF_2$ prism located in the bottom of a gas cell in the configuration described. Each spectrum consisted of 32 interferograms, recorded with a spectral resolution of 4 $cm^{-1}$. UV–Vis spectra were recorded on a HP-Hewlett Packard 8452 A spectrophotometer (Hewlett Packard, Palo Alto, CA, USA) in the 190–800 nm range.

### 2.1. Synthesis of Ligands (L1–L5)

The ligands were synthesized starting from L-proline. Five synthetic steps are necessary, according to the method reported in the literature [34–36] for the synthesis of L1. Some minor modifications were performed. Essentially, 205 mg (0.60 mmol) of ((S)-1-benzylpyrrolidin-2-yl)diphenylmethanamine (6) was added in a reaction flask containing 2.0 mL anhydrous methanol. Then, 1.05 eq of the salicylaldehyde derivative was added (63.94 μL (0.63 mmol) of 2-hydroxybenzaldehyde (L1), 96.00 mg (0.63 mmol) of 2-hydroxy-3-methoxybenzaldehyde (L2), 76.58 μL (0.63 mmol) of 2-hydroxy-3-methylbenzaldehyde (L3), 107.60 μL (0.63 mmol) of 3-tert-butyl-2-hydroxybenzaldehyde (L4), and 105.00 mg (0.63 mmol) of 3-ethoxy-2-hydroxybenzaldehyde (L5)), in addition to 20.00 mg (0.23 eq) of anhydrous $Na_2SO_4$(s). The reactions were left under stirring at 40 °C and followed by thin layer chromatography until no change in reagent consumption was observed. L1, L2, and L3 were filtered after 20 h of reaction and washed extensively with methanol to obtain yellowish solids. The solids were dissolved in dichloromethane and methanol was added to obtain yellowish crystals by slow evaporation for 24 h. L4 and L5 were left in reaction for 48 and 10 h, respectively, with the formation of a viscous yellowish solid. The solid was purified by column chromatography on silica gel with ethyl acetate/hexane (5:95) and (20:80) mixture as the eluent, respectively, for ligands L4 and L5.

(*E*)-2-((((1-benzylpyrrolidin-2-yl)diphenylmethyl)imino)methyl)phenol (L1). Yield: 75%. [1]HNMR (400 MHz, $CDCl_3$, 298 K): δ 14.71 (s, 1H), 7.97 (d, 2H), 7.33–7.20 (m, 7H), 7.18–7.01 (m, 8H), 6.96 (d, 1H), 6.75 (t, 1H), 3.96 (dd, 1H), 3.34 (d, 1H), 3.10 (d, 1H), 2.75–2.63 (m, 1H), 2.18–2.02 (m, 2H), 1.72–1.62 (m, 1H), 1.43–1.32 (m, 1H), 0.84–0.72 (m, 1H) ppm. [13]C NMR (400 MHz, CDCl3, 298 K): δ 164.55, 162.06, 144.55, 142.85, 140.44, 132.60, 132.09, 130.19, 129.10, 128.48, 128.12, 127.98, 127.83, 127.18, 126.86, 126.44, 118.87, 118.22, 117.51, 77.76, 71.96, 62.03, 55.10, 30.72, 24.00 ppm. HRMS (ESI$^+$, $CH_3OH$) *m/z* calculated for $C_{31}H_{31}N_2O$ 447.2436 $[M+H]^+$; found 447.2413. IR (KBr): 3347 (ν OH), 3056 (ν $Csp_2H$), 2969 (ν $Csp_3H$), 2818 (ν $Csp_3H$), 1620 (ν C=N), 1490 (ν C=C), 1280 (ν C–O) $cm^{-1}$. UV–Vis ε(L $mol^{-1}$ $cm^{-1}$) in $CH_2Cl_2$: 240 (8104), 260 (9513), 320 (3729), and 414 (540) nm.

(*E*)-2-((((1-benzylpyrrolidin-2-yl)diphenylmethyl)imino)methyl)-6-methoxyphenol (L2) Yield: 69%. $^1$HNMR (400 MHz, CDCl$_3$, 298 K): δ 15.23 (s, 1H), 7.91 (s, 1H), 7.45–7.40 (m, 2H), 7.32–7.20 (m, 6H), 7.17–7.00 (m, 7H), 6.82 (dd, 1H), 6.65–6.58 (m, 2H), 3.96 (dd, 1H), 3.31 (d, 1H), 3.08 (d, 1H), 2.76–2.69 (m, 1H), 2.18–2.06 (m, 2H), 1.72–1.63 (m, 1H), 1.39 (dd, 1H), 1.09 (dd, 1H) ppm. $^{13}$C NMR (400 MHz, CDCl$_3$, 298 K): δ 164.31, 155.95, 149.95, 144.01, 142.30, 139.91, 130.05, 128.87, 128.58, 128.30, 128.00, 127.95, 127.43, 127.00, 126.53, 123.74, 117.63, 116.63, 113.61, 77.22, 71.75, 62.00, 55.05, 30.70, 24.03 ppm. HRMS (ESI$^+$, CH$_3$OH) *m/z* calculated 477.2542 [M+H]$^+$; found 477.2518. IR (KBr): 3417 (ν OH), 3020 (ν Csp$_2$H), 2966 (ν Csp$_3$H), 2805 (ν Csp$_3$H), 1623 (ν C=N), 1491 (ν C=C), 1253 (ν C–O) cm$^{-1}$. UV–Vis (L mol$^{-1}$ cm$^{-1}$) in CH$_2$Cl$_2$: 232 (27,271), 266 (15,775), 324 (3046), and 432 (872) nm.

(*E*)-2-((((1-benzylpyrrolidin-2-yl)diphenylmethyl)imino)methyl)-6-methylphenol (L3). Yield: 72%. $^1$H NMR (400 MHz, CDCl$_3$, 298 K): δ 14.63 (s, 1H), 7.95 (s, 1H), 7.44 (dt, 2H), 7.32–7.02 (m, 14H), 6.87 (dd, 1H), 6.65 (t, 1H), 3.98 (dd, 1H), 3.49 (d, 1H), 3.15 (d, 1H), 2.64 (m, 1H), 2.26 (s, 3H), 2.15 (m, 1H), 2.07 (m, 1H), 1.72 (m, 1H), 1.35 (m, 1H), 0.97 (m, 1H) ppm. $^{13}$C NMR (400 MHz, CDCl$_3$, 298 K): δ 165.42, 160.18, 144.30, 143.29, 140.51, 133.45, 130.11, 129.78, 129.38, 128.57, 128.01, 127.73, 127.05, 126.79, 126.41, 126.28, 118.16, 117.74, 77.64, 72.13, 62.08, 54.98, 30.61, 23.89, 15.66 ppm. HRMS (ESI$^+$, CH$_3$OH) *m/z* calculated for C$_{32}$H$_{33}$N$_2$O461.2592 [M+H]$^+$; found 461.2569. IR (KBr): 3412 (ν OH), 3052 (ν Csp$_2$H), 2976 (ν Csp$_3$H), 2813 (ν Csp$_3$H), 1619 (ν C=N), 1491 (ν C=C), 1264 (ν C–O) cm$^{-1}$. UV–Vis ε(L mol$^{-1}$ cm$^{-1}$) in CH$_2$Cl$_2$: 232 (24,009), 262 (18,426), 326 (5085), and 420 (358) nm.

(*E*)-2-((((1-benzylpyrrolidin-2-yl)diphenylmethyl)imino)methyl)-6-(tert-butyl)phenol (L4). Yield: 45%. $^1$H NMR (400 MHz, CDCl$_3$, 298 K): δ 15.13 (s, 1H), 7.96 (s, 1H), 7.43 (dd, 2H), 7.35–7.06 (m, 14H), 6.88 (dd, 1H), 6.67 (dd, 1H), 3.96 (dd, 1H), 3.49 (d, 1H), 3.16 (d, 1H), 2.65 (ddd, 1H), 2.16–2.03 (m, 2H), 1.75–1.66 (m, 1H), 1.39 (s, 9H), 0.92–0.85 (m, 2H) ppm. $^{13}$C NMR (400 MHz, CDCl$_3$, 298 K): δ 165.45, 161.15, 144.44, 143.08, 140.63, 137.75, 130.28, 130.23, 129.40, 128.47, 127.89, 127.77, 126.99, 126.81, 126.33, 118.80, 117.30, 77.66, 72.23, 62.04, 54.97, 34.93, 30.64, 29.33, 23.81 ppm. HRMS (ESI$^+$, CH$_3$OH) *m/z* calculated for C$_{35}$H$_{39}$N$_2$O 503.3062 [M+H]$^+$; found 503.3035. IR (KBr): 3413 (ν OH), 3058 (ν Csp$_2$H), 2954 (ν Csp$_3$H), 1619 (νC=N), 1264 (ν C–O) cm$^{-1}$. UV–Vis ε(L mol$^{-1}$ cm$^{-1}$) in CH$_2$Cl$_2$: 232 (22,025), 264 (16,192), 330 (5320), and 400 (269) nm.

(*E*)-2-((((1-benzylpyrrolidin-2-yl)diphenylmethyl)imino)methyl)-6 ethoxy phenol (L5). Yield: 38%. $^1$H NMR (400 MHz, CDCl$_3$, 298 K): δ 14.92 (s, 1H), 7.93 (s, 1H), 7.44 (dd, 2H), 7.32–7.05 (m, 13H), 7.01 (dd, 2H), 6.84 (t, 1H), 6.63 (d, 2H), 4.08 (q, 2H), 3.99 (dd, 1H), 3.42 (d, 1H), 3.12 (d, 1H), 2.66 (m, 1H), 2.10 (m, 2H), 1.70 (m, 1H), 1.45 (t, 3H), 1.35 (m, 1H), 1.03 (m, 1H) ppm. $^{13}$C NMR (400 MHz, CDCl$_3$, 298 K): δ 165.11, 154.02, 148.19, 144.16, 143.05, 140.33, 132.44, 130.08, 130.00, 129.19, 128.59, 128.15, 127.96, 127.79, 127.18, 126.85, 126.45, 123.76, 118.48, 117.15, 115.08, 77.50, 71.86, 64.38, 62.06, 54.99, 30.63, 23.95, 14.88 ppm. HRMS (ESI$^+$, CH$_3$OH) *m/z* calculated for C$_{32}$H$_{33}$N$_2$O$_2$ 477.2537 [M+H]$^+$; found 477.2503. IR (KBr): 3421 (ν OH), 3055 (νCsp$_2$H), 2965 (ν Csp$_3$H), 1621 (ν C=N), 1492 (ν C=C), 1272 (ν C–O) cm$^{-1}$. UV–Vis ε (L mol$^{-1}$ cm$^{-1}$) in CH$_2$Cl$_2$: 232 (15,984), 264 (9288), 332 (2716), and 430 (622) nm.

## 2.2. General Procedure of Synthesis of Cu$^{II}$ Chlorido Complexes

In a reaction flask, 50.00 mg of CuCl$_2$ (0.37 mmol, 1.1 eq) wasadded to 3.0 mL of anhydrous methanol. The methanolic solution was heated at reflux temperature for 10 min, followed by the addition of 1.0 eq of the ligands (150.00 mg (0.33 mmol) of HL1, 157.00 mg (0.33 mmol) of HL2, 152.00 mg (0.33 mmol) of HL3, 166.00 mg (0.33 mmol) of HL4 and 162.00 mg (0.33 mmol) of HL5). After 4 h, the reaction mixture was cooled to room temperature and filtered. The filtrate was evaporated to dryness and suspended in dichloromethane. The mixture was centrifuged and the supernatant was removed. The complexes were obtained after removal of the solvents by rotatory evaporation under vacuum.

Spectroscopic data of Cu$^{II}$L1. Dark green powder, yield 75%. HRMS (ESI$^+$, CH$_2$Cl$_2$/CH$_3$CN) *m/z* 566.1162 calculated for [M+Na]$^+$, found 566.1122; IR (KBr): 3546, 3472, 3412 (ν OH), 3080, 3057 (ν Csp$_2$H), 3026 (ν N=Csp$_2$H), 2959, 2922, 2851 (ν Csp$_3$H), 1654, 1637, 1617(ν C=N), 1598, 1580 (ν C=C), 1455, 1445 (δ CH2), 1317, 1276, 1261 (ν C–O), 1089, 1074, 1028 (ν C–N), 760, 704 (γ Csp$_2$H), 638 (ν Cu–O),

474 ($\nu$ Cu–N) cm$^{-1}$. UV–Vis Vis $\varepsilon$ L mol$^{-1}$ cm$^{-1}$ (CH$_2$Cl$_2$): 248 (18,246), 276 (16,002), 380 (4383), 636 (265) nm. C$_{31.5}$H$_{30.5}$C$_{l1.5}$CuN$_2$O$_{1.25}$[Cu(L1)Cl]·(0.25CH$_3$OH)·(0.25CH$_2$Cl$_2$) calculated C, 65.93; H, 5.36; N, 4.88. Found: C, 65.95; H, 5.23; N, 4.95.

Spectroscopic data of Cu$^{II}$L2. Dark brown powder, yield 91%. HRMS (ESI$^+$, CH$_2$Cl$_2$/CH$_3$CN) *m/z* calculated for [M–Cl]$^+$ 538.1676, found 538.1669; HRMS (ESI$^+$, CH$_2$Cl$_2$/CH$_3$OH) *m/z* calculated 1169.2638 for [2M + Na$^+$]$^+$, found *m/z* 1169.2410; IR (KBr): 3458, 3410 ($\nu$ OH), 3055 ($\nu$ Csp$_2$H), 3026 ($\nu$ N=Csp$_2$H), 2959, 2926 ($\nu$ Csp$_3$H), 1619 ($\nu$ C=N), 1577, 1544 ($\nu$ C=C), 1469, 1444 ($\delta$ CH2), 1316, 1276 ($\nu$ C–O), 1245, 1218 ($\nu$ C–O–C), 1081, 1004 ($\nu$ C–N), 748, 704 ($\gamma$ Csp$_2$H), 638 ($\nu$ Cu–O), 557 ($\nu$ C–N) cm$^{-1}$. UV–Vis Vis $\varepsilon$(L mol$^{-1}$ cm$^{-1}$) in CH$_2$Cl$_2$: 234 (15,660), 284 (13,953), 362 (2479), ~600 (–) nm. C$_{32.6}$H$_{32.2}$Cl$_{2.2}$CuN$_2$O$_2$[Cu(L2)Cl]·(0.66CH$_2$Cl$_2$) calculated C, 62.16; H, 5.16; N, 4.44. Found: C, 62.14; H, 4.82; N, 4.71

Spectroscopic data of Cu$^{II}$L3. Green brownish powder, yield 91%. HRMS (ESI$^+$, CH$_2$Cl$_2$/CH$_3$OH) *m/z* calculated for [M–HCl]$^+$ 522.1727, found 522.1691. IR (KBr): 3549, 3450, 3410 ($\nu$ OH), 3082, 3057 ($\nu$ Csp$_2$H), 3026 ($\nu$ N=Csp$_2$H), 2949, 2920 ($\nu$ Csp$_3$H), 1654, 1615 ($\nu$ C=N), 1600, 1577, 1544 ($\nu$ C=C), 1467, 1446, 1421 ($\delta$ CH2), 1317, 1276 ($\nu$ C–O), 1087, 1028 ($\nu$ C–N), 748, 704 ($\gamma$ Csp$^2$H), 638 ($\nu$ Cu–O), 567 ($\nu$Cu–N) cm$^{-1}$. UV–Vis Vis $\varepsilon$(L mol$^{-1}$ cm$^{-1}$) in CH$_2$Cl$_2$:252 (19,765), 280 (12,308), 378 (3053), 600–700 (–) nm. C$_{33}$H$_{34}$Cl$_2$CuN$_2$O$_{1.5}$ [Cu(L3)Cl]·(0.5CH$_3$OH)·(0.5CH$_2$Cl$_2$) calculated C, 64.23; H, 5.55; N, 4.54. Found: C, 64.00; H, 5.78; N, 4.48.

Spectroscopic data of Cu$^{II}$L4. Dark green, yield 72%. HRMS (ESI$^+$, CH$_2$Cl$_2$/CH$_3$OH) *m/z* calculated for [M–HCl]$^+$ 564.2196, found 564.2168. IR (KBr): 3545, 3472, 3412 ($\nu$OH), 3084, 3054 ($\nu$ Csp$^2$H), 3026 ($\nu$ N=Csp$^2$H), 2949, 2920 ($\nu$ Csp$^3$H), 1654, 1615 ($\nu$ C=N), 1596, 1534, 1492 ($\nu$ C=C), 1465, 1443, 1415 ($\delta$ CH$_2$), 1336, 1326 ($\nu$ C–O), 1143, 1085 ($\nu$ C–N), 748, 702 ($\gamma$ Csp$^2$H), 567 ($\nu$ Cu–N) cm$^{-1}$. UV–Vis Vis $\varepsilon$(L mol$^{-1}$ cm$^{-1}$) in CH$_2$Cl$_2$:250 (18,922), 278 (12,694), 332 (3959), 388 (4499) e 650 (294) nm. C$_{36}$H$_{39.8}$Cl$_{2.2}$CuN$_2$O$_{1.4}$[Cu(L4)Cl]·(0.4CH$_3$OH)·(0.6CH$_2$Cl$_2$) calculated C, 65.07; H, 6.04; N, 4.22. Found: C, 64.84; H, 5.97; N, 4.71.

Spectroscopic data of Cu$^{II}$L5. Dark brown, yield 96%. HRMS (ESI$^+$, CH$_2$Cl$_2$/CH$_3$OH) *m/z* calculated for [M–HCl]$^+$ 552.1833, found 552.1784; [M–CH$_4$–HCl]$^+$ 536.1519, found 536.1691. IR (KBr): 3458, 3414($\nu$ OH), 3080, 3057 ($\nu$ Csp$^2$H), 3026 ($\nu$ N=Csp$^2$H), 2974, 2924, 2853 ($\nu$ Csp$^3$H), 1654, 1615 ($\nu$ C=N), 1602, 1577, 1560 ($\nu$ C=C), 1465, 1448 ($\delta$ CH$_2$), 1317, 1278 ($\nu$ C–O), 1245, 1216 ($\nu$ C–O–C), 1073, 1028 ($\nu$ C–N), 763, 740, 704 ($\gamma$ Csp$^2$H), 638 ($\nu$ Cu–O) cm$^{-1}$. UV–Vis Vis $\varepsilon$(L mol$^{-1}$ cm$^{-1}$) in CH$_2$Cl$_2$: 236 (10,803), 252 (15,770) 356 (2348), ~600 (n.d.) nm. C$_{34.1}$H$_{35.7}$Cl$_{2.7}$CuN$_2$O$_{2.25}$ [2Cu(L5)Cl]·(1.6CH$_2$Cl$_2$) calculated C, 61.57; H, 5.21; N, 4.31. Found: C, 61.64; H, 4.87; N, 4.44.

## 2.3. Synthesis of Cu$^{II}$Perchlorate Complex

The perchlorate complex was synthesized similarly to the chlorido complex, but using the precursor Cu(ClO$_4$)·6H$_2$O. After 4 h of reaction, cold distilled water was added and the obtained solid was centrifuged, filtered, and washed with cold distilled water. The solid was left in a desiccator at high vacuum. Dark green. Yield: 89%. HRMS (ESI$^+$, CH$_2$Cl$_2$/CH$_3$OH) *m/z* calculated for [2M–ClO$_4$$^-$]$^+$ 1175.2848, found 1175.2834. IR (KBr): 3530, 3446, 3317 ($\nu$ OH), 3060 ($\nu$ Csp$^2$H), 3033 ($\nu$ N=Csp$^2$H), 2967, 2841 ($\nu$ Csp$^3$H), 1655, 1623 ($\nu$ C=N), 1606, 1577, 1545 ($\nu$ C=C), 1493, 1470, 1440 ($\delta$ CH$_2$), 1319, 1279 ($\nu$ C–O), 1246, 1220 ($\nu$ C–O–C), 1120, 1108, 1087 ($\nu_3$ ClO$_4$$^-$), 1005 ($\nu$ C–N), 943, 921 ($\nu_4$ ClO$_4$$^-$) 748, 706 ($\gamma$ Csp$^2$H), 638, 624 ($\nu$ Cu–O), 556, 522 ($\nu$ Cu–N)cm$^{-1}$. UV–Vis (CH$_2$Cl$_2$): 244, 286, 392, 592 nm. C$_{65.3}$H$_{66.6}$Cl$_{2.6}$Cu$_2$N$_4$O$_{13}$ [2Cu$^{II}$L2ClO$_4$·(CH$_3$OH)](0.3CH$_2$Cl$_2$) calculated C, 58.76; H, 5.03; N, 4.20. Found: C, 58.87; H, 5.15; N, 4.16. Conductivity: 13 $\mu$S cm$^{-1}$ in dichloromethane.

## 2.4. Catalysis Protocol

For the catalytic assays, fresh solutions of urea (60 mM in water) and of the complexes (1.6 mM in dichloromethane (DCM)) were prepared.

The catalytic assays were performed by diluting 50 $\mu$L of the solution containing the complex into 850 $\mu$L of the reaction solvent (methanol, acetonitrile, and dimethylsulfoxide, tetrahydrofuran,

and ethanol). The reaction typically started by the addition of the urea solution (100 μL). Vigorous stirring was maintained over the course of the reaction. Aliquots of 100 μL of the reaction were taken after 5, 10, 20, 30, 40, 60, 120, 240 and 480 s of reaction. These aliquots were analyzed by the Berthelot method to quantify the ammonia content. The reaction temperature was kept at 36 °C. All reactions were performed in triplicate.

For each of the catalytic protocols, a control experiment was also performed without the complexes and the blank was subtracted from the catalytic measurements.

The amount of water in the medium, when increasing amount of urea concentrations were employed, were 10, 20, 30 and 40%, respectively, to the urea concentrations of 5.2, 10.4, 15.6 and 20.4 mM (except in acetonitrile, in which urea 20.4 mM was also employed).

*2.5. Ammonia Quantification*

Fresh solutions of sodium hypochlorite (2.5% in water), sodium citrate (0.38 mM in 0.46 mM aqueous solution of NaOH), and sodium salicylate (2.75 mM in water containing 9.39 μmol of sodium nitroprussiate) were prepared prior to the ammonia quantification. To a microtube containing 250 μL of the hypochlorite solution and 250 μL of citrate/NaOH solution, the 100 μL aliquot from the reaction was added, followed by the addition of 300 μL of the sodium salicylate solution. After mixing, the reaction was allowed to proceed for 15 min at room temperature. The reaction was analyzed by UV–Vis spectroscopy at 654 nm. Quantification of ammonia was performed by using a calibration curve with seven measurements and $R^2 = 0.99$, using ammonium chloride as standard.

*2.6. Computer Simulations*

Geometries for the $Cu^{II}L1$ monomer and the dimer were optimized at the semi-empirical GFN1-xTB level in vacuum using the xtb software [37–39], with convergence criteria for the self-consistent charge (SCC) iterations of $2·10^{-7}$ $E_h$ for the energy change and $4·10^{-6}$ for the charge change between cycles and for the geometry optimization of $1·10^{-6}$ $E_h$ for the energy change between steps and a maximum gradient of $8·10^{-4}$ $E_h/α$. All calculations were performed with an electronic temperature of 500 K to allow some degree of Fermi smearing of nearly-degenerate energy levels and to take static correlation into account. All model systems were considered neutral with each Cu(II) cation in the $Ar[3d^9]$ configuration, yielding a doublet state for the monomer. Regarding the dimer, both singlet and triplet states were considered for the sake of completeness. The same protocols were applied to acetonitrile, water, and methanol molecules in vacuum. After full geometry optimization of the monomer and the two spin states of the dimer, further geometric relaxation was performed by means of 10 ps-long molecular dynamics simulations performed in the canonical, constant-NVT ensemble (meaning amount of substance (N), volume (V) and temperature (T) are conserved), using the Berendsen weak-coupling scheme to control the temperature around 300 K. Equations of motion were integrated using a 0.5 fs time step, and recording structures and energies each of 0.5 ps. The last structure for each system was subjected to further geometry optimization and these optimized structures were considered as the lowest lying reference states for the thermochemical analyses. The next step of the modeling consisted of searching the thermodynamically most probable position and relative orientation of each solvent around each complex, as described in detail in the Supplementary Materials. This systematic search amounted to ca. 1 million quantum chemical calculations and the most probable solvent-$Cu^{II}L1$ structures were subjected to further geometry optimization as described above.

## 3. Results and Discussion

*3.1. Probing Copper Coordination*

Amine 6 was obtained after a series of reactions starting from *L*-proline (Figure 2). The reaction between amine 6 with the corresponding aldehyde afforded ligands L1–L5. After purification, these compounds were characterized by $^1$H NMR, $^{13}$C NMR, 2D NMR, mass spectrometry, infrared

spectroscopy, and X-ray diffractometry (see support information). The coordination of L1–L5 to CuCl$_2$ in methanol resulted in the formation of complexes Cu$^{II}$L1–L5 in high yields (71–91%). All complexes were characterized by proper microanalysis, EPR, infrared spectroscopy, mass spectrometry, cyclic voltammetry, and electronic spectroscopy at the UV–Vis region (Supplementary Materials). Some features of the characterization of the complexes deserve to be described since they indicate the coordination mode and the extent of dimerization. For instance, through FTIR spectroscopy (Figures S25–S29), it was evident that the coordination occurred via the nitrogen of azomethine [40,41], since the ν C=N vibration mode was red shifted by 4–7 cm$^{-1}$. An increase of the νC–O energy indicated that the coordination was also occurring via a phenolate [34]. Hence, the presence of a band in the 640–470 cm$^{-1}$ range corroborated that coordination occurred through oxygen and nitrogen, consistent with the ν M–O and M–N modes [42].

**Figure 2.** Synthesis of L1–L5 ligands from L-proline. Reaction conditions: (**a**) SOCl$_2$(l), CH$_3$OH; (**b**) BrCH$_2$(Ph), K$_2$CO$_3$(s), CH$_3$CN; (**c**) MgBr(Ph), THF; (**d**) NaN$_3$(s), H$_2$SO$_4$ 70%, CHCl$_3$; (**e**) LiAlH$_4$(s), THF; (**f**) salicylaldehyde and derivatives, Na$_2$SO$_4$(s), CH$_3$OH.

Corroborating to the FTIR spectroscopy, a bathocromic shift was observed for the π→π* bands of azomethine in the electronic spectroscopy, characterizing the coordination through this moiety [43]. In addition, the disappearance of the n→π* azomethine transition indicated that the Cu$^{II}$ coordination occurred on that position of the Schiff base [40]. Moreover, a broadband in the 550–770 nm region was consistent with the d–d transition of the metal center [43–45] of complexes Cu$^{II}$L1 and Cu$^{II}$L4 (636 and 650 nm, respectively). However, the other complexes exhibited a less evident d–d band, possibly due to a higher planar geometry in comparison to Cu$^{II}$L1 and Cu$^{II}$L4 (Figures S30–S33) [46].

In the cyclic voltammetry, for all ligands, the oxidation of phenol to quinone and the formation of radical cations was observed. These observations are in agreement with the redox behavior of other Schiff bases [47,48]. After coordination, these redox processes were displaced to higher potentials as an effect of electron depletion upon coordination by the metal [49].

The possibility of the formation of monomeric and dimeric structures was first inspected by high resolution mass spectrometry (HRMS) in which, for example, a peak corresponding to the monomer was present at *m/z* 538.1669, whereas the dimer was observed at *m/z* 1169.2410 for the Cu$^{II}$L2 complex (Figure S58). However, since dimers can be formed in the gas phase depending on the solution concentration [50], we performed EPR analysis in solid and in solution. With these analyses, we evaluated the existence of equilibrium between monomers and dimers in solution that could be controlled by ligand substitution reactions.

*3.2. Ligand Substitution and Electron Paramagnetic Resonance Measurements*

First, ligand substitution was followed by conductivity measurements to give us insights on the natural dissociation of the complexes. In dichloromethane (DCM) at 298 K, the conductivities were consistent with neutral compounds (values in the 2.00–6.00 μS cm$^{-1}$ range). However, in acetonitrile (ACN) at 298 K, the conductivity was shown to constantly increase over time, somewhat reaching a plateau after half an hour (Figure 3A). The quasi-stabilized values for complexes Cu$^{II}$L2, CuIIL3,

and Cu$^{II}$L4 were 63, 43, and 23 µS cm$^{-1}$, respectively. These values indicate that the labilization of the chlorido ligand by acetonitrile was more pronounced for Cu$^{II}$L2. The solvolysis of the chlorido ligand was facilitated when water was present in solution, as shown in Figure 3B, which shows that in 2 min, the conductivity was already stabilized in limiting values of the 1:1 electrolyte range (55.0–90.0 µS cm$^{-1}$) [51]. Interestingly, the conductivity of these complexes was more stable both in methanol and the methanol/water mixture (80/20 *v/v*), revealing a slower rate of ligand substitution in these solvents (Figures S53–S55). The observed values ranged from 45.0 to 53.0 µS cm$^{-1}$ and 47.0 to 50.0 µS cm$^{-1}$ in methanol and the methanol/water mixture, respectively, which were lower than that expected for the 1:1 electrolyte in methanol. The lower values might indicate the presence of a mixture of charged and neutral species in solution. The displaced chloride could be detected by the addition of silver nitrate solution as a white precipitate of AgCl.

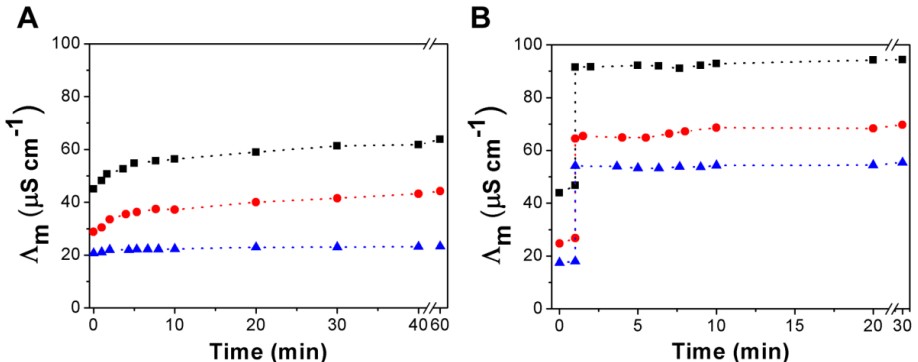

**Figure 3.** Influence of the time in conductivity measurements of the complexes in (**A**) acetonitrile and (**B**) acetonitrile/water (80/20) mixture. Conductivity of Cu$^{II}$L2 is shown as black squares, Cu$^{II}$L3 is shown as red circles, and Cu$^{II}$L4 is shown as blue triangles.

Considering the existence of a mixture of species in solution, EPR measurements were performed in the same solvents of the conductivity analyses to enable a better structural comprehension (Figure 4). In dichloromethane, all complexes presented values of $g_z$ higher than $g_x$ and $g_y$ (Table 1 and Table S7), suggestive of an axial symmetry [52–54], due to the presence of an unpaired electron in the $dx^2-y^2$ orbital. The axial symmetry supports the proposal of a square planar geometry of the complexes [42,55,56]. However, as shown in Table 1, the differences between the values of $g_x$ and $g_y$ are indicative of a distortion of the plane. In addition, the highest observed $g_z$ for Cu$^{II}$L4 could mean that this complex has a tetrahedral distortion. In contrast, the $A_z$ is smaller for the Cu$^{II}$L5 complex as a result of a greater distortion, indicating that aside from the difference between substituents in the ligands, another factor might be affecting the tetrahedral distortion of complexes Cu$^{II}$L4 and Cu$^{II}$L5.

**Table 1.** Electron Paramagnetic Resonance parameters for the Cu$^{II}$ complexes of this work in dichloromethane at 298 K.

| Compounds | G | | | $t_{corr}$, ps | A, cm$^{-1}$ ($\times 10^{-4}$) | | |
|---|---|---|---|---|---|---|---|
| | $g_x$ | $g_y$ | $g_z$ | | $A_x$ | $A_y$ | $A_z$ |
| Cu$^{II}$L1 | 2.0501 | 2.0932 | 2.1626 | 71.9 | 14.53 | 11.51 | 200.2 |
| Cu$^{II}$L2 | 2.0584 | 2.0932 | 2.1626 | 114.5 | 11.88 | 11.51 | 198.8 |
| Cu$^{II}$L3 | 2.0515 | 2.0932 | 2.1626 | 75.3 | 11.88 | 11.51 | 201.0 |
| Cu$^{II}$L4 | 2.0494 | 2.0518 | 2.2063 | 44.2 | 14.53 | 14.06 | 199.5 |
| Cu$^{II}$L5 | 2.0504 | 2.0932 | 2.1626 | 116.7 | 11.88 | 11.51 | 188.6 |

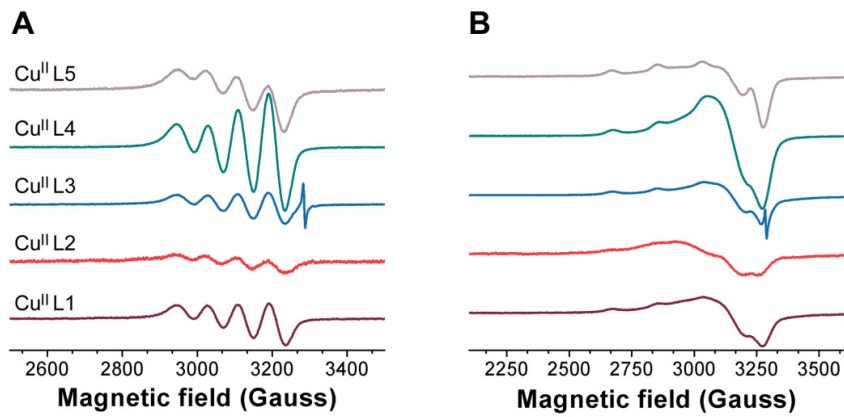

**Figure 4.** Comparison of the experimental EPR spectra of the complexes in dichloromethane at (**A**) 298 and (**B**) 77 K. Complex $Cu^{II}L1$ is shown in purple, $Cu^{II}L2$ is shown in red, $Cu^{II}L3$ is shown in blue, $Cu^{II}L4$ is shown in green, and $Cu^{II}L4$ is shown in grey lines. The narrow line in the high magnetic field for $Cu^{II}L3$ are a standard signal of $Cr^{III}$:MgO sample (g = 1.9797).

In acetonitrile and acetonitrile/water (80:20) mixture, the values of $g_x$ and $g_y$ were more similar to each other, as an effect of ligand substitution, in which the nitrogen atom of ACN or the oxygen atom of water are coordinated to copper, forming a plane with higher symmetry than the previous N, N, O, Cl coordination plane. The increase in symmetry observed in the EPR measurements in ACN and ACN/$H_2O$ is in agreement with the chlorido displacement observed in the conductivity measurements. However, there was still no evidence of dimeric and monomeric species in equilibrium in these solutions.

Hence, the spinning radiuses of the molecules in solution were obtained from the spectral simulations using the EasySpin program [57] to obtain the rotational time correlation $t_{corr}$ (Table 2). These values are associated withthe spinning velocity of a molecule in solution, expressing large values of $t_{corr}$ when effective intramolecular interactions are generated. Curiously, the increasing order of $t_{corr}$ was $Cu^{II}L4 > Cu^{II}L1 > Cu^{II}L3 > Cu^{II}L2 > Cu^{II}L5$, revealing that complexes bearing a substituent group with oxygen ($Cu^{II}L2$ and $Cu^{II}L5$) are more effectively interacting in solution than $Cu^{II}L4$. With the $t_{corr}$ values, we were able to calculate the radiuses and volumes of rotation of the complexes using the Stokes–Einstein–Debye (SED) equation ($t_{corr} = 4\pi\eta a^3/3K_BT$), where a is the molecular radius of rotation (Table 2). In general, all compounds have reduced their radius of rotation in acetonitrile and the acetonitrile/water (80:20) mixture. Therefore, the complexes were probably arranged mostly as dimeric structures in dichloromethane, and when coordinating solvents were present such as acetonitrile and water, the equilibrium between dimeric and monomeric species shifted to monomeric ones.

**Table 2.** Comparison of the values of the radiuses of rotation and approximate volumes of the complexes $Cu^{II}L1$–$Cu^{II}L5$ considering a spherical model.

| Compounds | a, Å | | | V, Å³ (×10²) | | |
|---|---|---|---|---|---|---|
| | DCM | ACN | ACN/$H_2O$ | DCM | ACN | ACN/$H_2O$ |
| $Cu^{II}L1$ | 5.55 | 4.20 | 4.38 | 7.16 | 3.10 | 3.51 |
| $Cu^{II}L2$ | 6.49 | 5.14 | 4.19 | 11.4 | 5.68 | 3.08 |
| $Cu^{II}L3$ | 5.64 | 4.69 | 4.14 | 7.51 | 4.32 | 2.97 |
| $Cu^{II}L4$ | 4.72 | 4.69 | 4.34 | 4.40 | 4.32 | 3.42 |
| $Cu^{II}L5$ | 6.53 | 4.68 | 4.48 | 11.7 | 4.29 | 3.76 |

In fact, when frozen solutions of the complexes in DCM were analyzed by EPR spectra and compared to simulated spectra (Figure 5), it was evident that a component attributed to molecular aggregates needed to be introduced for a better fit. These molecular aggregates had a magnetic

interaction, indicating that two Cu$^{II}$ centers probably interacted with each other, causing the lines to broaden. This broadening was more pronounced for Cu$^{II}$L2 and Cu$^{II}$L5, which were the most effective complexes to form dimeric structures, as observed by the t$_{corr}$ values. A similar trend was observed for the measurements performed in acetonitrile and in the acetonitrile/water (80:20) mixture, indicating equilibrium between monomeric and dimeric species in frozen solution. The stronger interaction observed for complexes Cu$^{II}$L2 and Cu$^{II}$L5 might indicate the presence of halogen-bonds between the oxygen from methoxy and ethoxy radicals with the chloride. Halogen bonds are more sensitive to steric effects and could be the reason for the higher volume observed for Cu$^{II}$L2 in ACN than Cu$^{II}$L5 [58]. Moreover, the dimerization of copper complexes in different solvents has already been observed by EPR measurements by other groups [56], corroborating our observations.

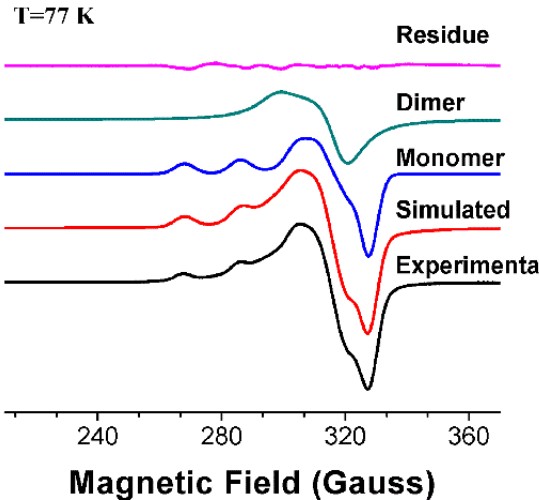

**Figure 5.** Comparison between the experimental and simulated EPR spectrum of Cu$^{II}$L4 complex at 77 K in dichloromethane. The experimental EPR spectrum is shown as a black line, whereas the simulated spectrum is shown as a red line. Simulation spectra are composed by a monomeric and dimeric species that are shown in blue and green, respectively.

The difference in the g$_z$ observed for the different complexes may be a result of distinct aggregation structures. For instance, Cu$^{II}$L1, Cu$^{II}$L2, and Cu$^{II}$L5 present values of g$_z$0 higher than g$_x$0 and g$_y$0, which are similar to monomeric species, possibly due to the maintenance of an axial geometry even after aggregation. An opposite behavior was observed for complexes Cu$^{II}$L3 and Cu$^{II}$L4, which hadg$_z$0 values lower than g$_x$0 and g$_y$0 in the aggregate species. Therefore, these complexes (Cu$^{II}$L3 and Cu$^{II}$L4) may have formed aggregates with nonaxial geometry, suggesting that the unpaired Cu$^{II}$ electron is not of the d$x^2$–y$^2$ orbital, possibly due to a trigonal bipyramidal geometry, as shown in Figure 6. Structural features behind thermodynamic differences for the interactions between one solvent molecule and the Cu$^{II}$L1 monomer and dimer were evaluated by quantum chemical calculations. The dimers had very distinctive geometries in each electronic spin state, with the lowest-lying singlet states having a single Cl bridge between the two monomers (Figure 6B–D), which renders each monomer structurally different from the other, while the higher energy triplet state has two Cl bridging the two Cu(II) atoms (Figure S76). It was evidenced by the simulations that for all of the solvents considered as well as for the bare dimer in vacuum, the singlet state was always lower lying than the triplet state, corroborating the proposition of the structures in Figure 6.

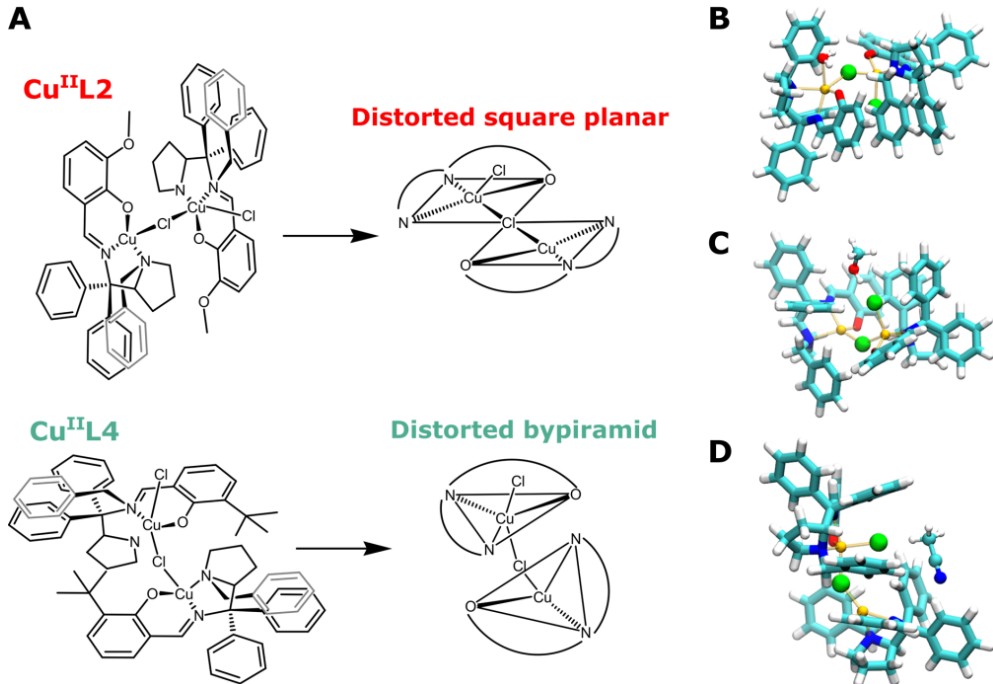

**Figure 6.** Proposition of the aggregate structures of complexes Cu$^{II}$L2 and Cu$^{II}$L4 in frozen solutions of dichloromethane (**A**). Optimized structures for the Cu$^{II}$L1dimer in its singlet state interacting with H$_2$O (**B**), MeOH (**C**), and ACN (**D**).

Since all complexes behaved similarly in the EPR measurements in dichloromethane, acetonitrile, and the acetonitrile/water mixture at 298 K, only Cu$^{II}$L2 was evaluated in methanol and the methanol/water in EPR measurements at 298 K. To provide a comparison between dimeric and monomeric species in solution, an analogous of complex Cu$^{II}$L2 was synthesized with perchlorate as a counterion ([Cu$^{II}$L2ClO$_4$] Figure 1). The perchlorate ion is known for its high volume and would be expected to generate only monomeric species in solution. Therefore, this complex was compared with Cu$^{II}$L2 in EPR measurements performed at 298 K in methanol and the methanol/water (80/20, *v/v*) mixture. Interestingly, the EPR spectrum of Cu$^{II}$L2 in methanol (Figure 7) is visibly a mixture between two species. However, unexpectedly, [Cu$^{II}$L2ClO$_4$], despite presenting a profile of monomeric species in solution, exhibited a three times higher t$_{corr}$ than Cu$^{II}$L2, which could indicate that [Cu$^{II}$L2ClO$_4$] is in fact, dimeric. Indeed, the HRMS spectra exhibited *m/z* peaks corresponding to dimeric structures (1175.2834, Figure S59) and in the FTIR spectra, three bands associated with monodentated ClO$_4^-$ species were observed at 1121, 1108 and 1027 cm$^{-1}$, corroborating the hypothesis of [Cu$^{II}$L2ClO$_4$] complex dimerization. The addition of 20% water keeps the equilibrium in solution, as expected, due to the similarity of the conductivities of the complexes in methanol and the methanol/water mixtures. Therefore, it can be assumed that methanol and water do not fully displace the chloride. The unexpected dimerization of [Cu$^{II}$L2ClO$_4$] might strengthen the proposition of halogen bond formation in the Cu$^{II}$L2 complex, suggesting a supramolecular structure with the possibility of use in dynamic catalysis [59].

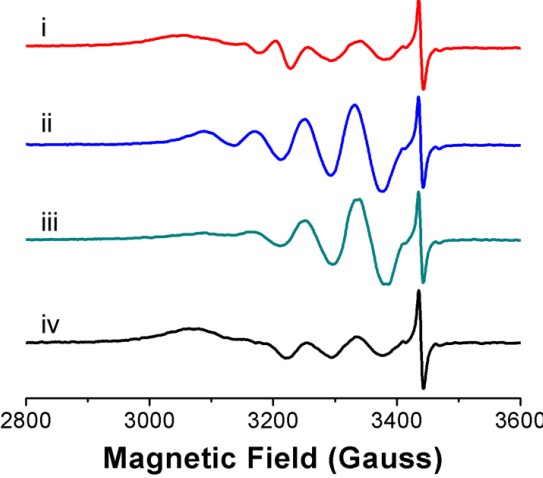

**Figure 7.** EPR spectra of Cu$^{II}$L2 and [Cu$^{II}$L2ClO$_4$] in methanol (i and ii) and Cu$^{II}$L2 and [Cu$^{II}$L2(ClO$_4$)]in the methanol/water (80:20) mixture (iii and iv). The narrow line in the high magnetic field for Cu$^{II}$L3 isa standard signal of the Cr$^{III}$:MgO sample (g = 1.9797).

### 3.3. Urea Hydrolysis as a Model Reaction: Kinetics of NH$_3$ Formation

The self-organization of these complexes into dimeric or monomeric structures was shown to be dependent on the solvent and ligand exchange reactions, as shown in Figure 8. Due to the distinct behaviors of the complexes in acetonitrile/water and methanol/water, we suspected that hydrolytic catalysis could be tuned by influencing the equilibrium between monomeric and dimeric species. Hence, we decided to evaluate their potential as catalysts to hydrolyze urea, as a model reaction. Catalysis was performed primarily in the acetonitrile/water and methanol/water mixtures. In this reaction, ammonia is expected to be formed and aliquots of the reaction were analyzed by the Berthelot method [60] over the reaction times (5, 10, 20, 30, 40, 50, 60, 120, 240, and 480 s).

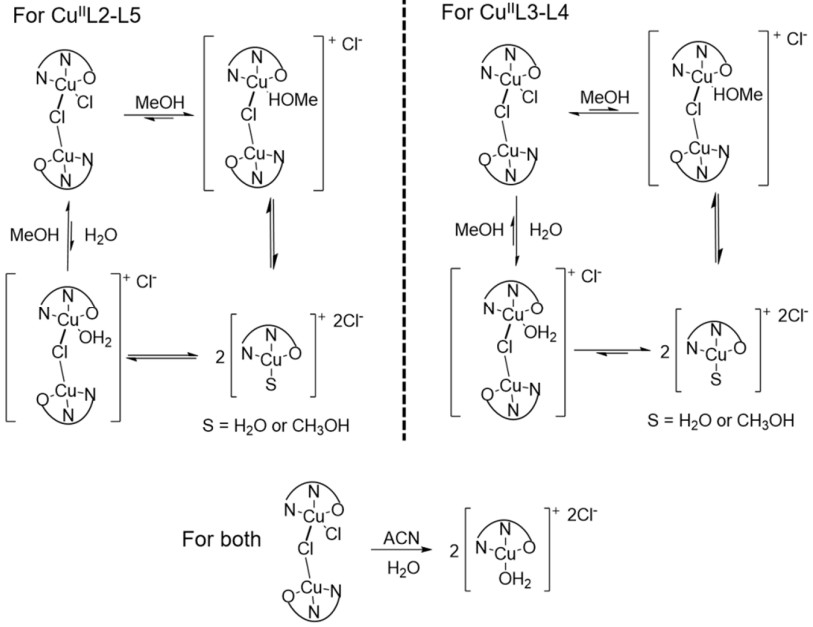

**Figure 8.** Equilibrium of dimeric and monomeric species dependent on water-association solvent.

Complexes Cu$^{II}$L2, Cu$^{II}$L3, and Cu$^{II}$L4 were chosen to evaluate the influence of the complexes' aggregation in the reaction. In the acetonitrile/water mixture, the reaction was faster than the employed

method to detect ammonia formation (minimum reaction time: 5 s) and we could only observe the decrease in ammonia concentration over the reaction time (Figure 9A and Figures S63–S66). In contrast, in the methanol/water mixtures, the reaction profile changed, in which the increase in ammonia concentration was observed until a saturation level was reached (Figure 9B). The lability of the chlorido was smaller in methanol and the methanol/water (80/20) mixture when compared to the acetonitrile system, and indicates that the labilization of chlorido affects the path of the reaction. For instance, in acetonitrile/water, all complexes exhibited a decrease of the volume, possibly due to the formation of monomeric species in solution. Therefore, the equilibrium dimer/monomer is still present in methanol/water and it can be inferred that the presence of dimeric structures in solution is possibly slowing the reaction.

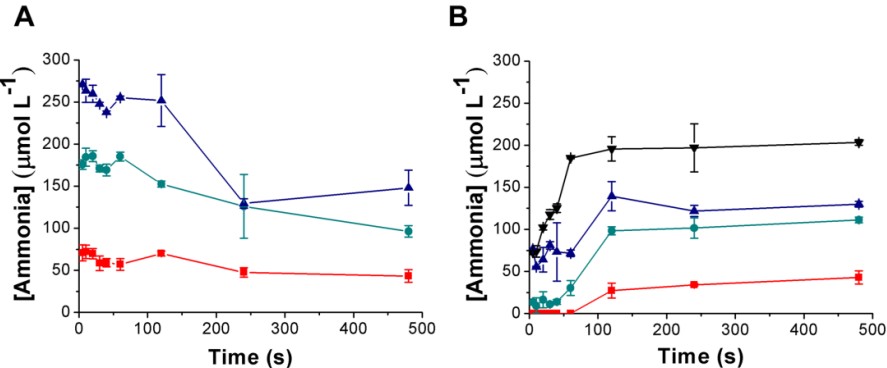

**Figure 9.** Ammonia quantification produced by $Cu^{II}L2$ complex in (**A**) acetonitrile/water and (**B**) methanol/water mixtures up to 480 s at 308 K and at different urea concentrations: 5.2 mmol $L^{-1}$ (brown line, squares), 10.4 mmol $L^{-1}$ (green line, circles), 15.6 mmol $L^{-1}$ (blue line, triangles), and 20.8 mmol $L^{-1}$ (black line, inverted triangles).

Considering that complex [$Cu^{II}L2ClO_4$] was mostly dimeric in solution, it would be expected to observe a slower reaction rate of urea hydrolysis by this complex. In general, the behavior of [$Cu^{II}L2ClO_4$] was similar to the chloride complex (faster in acetonitrile/water and slower in methanol/water mixtures), but indeed, a four times lower conversion was observed (Figure 10). Moreover, the reaction in methanol/water only started to produce ammonia after 5 min of reaction, whereas the $Cu^{II}L2$ complex was able to produce it after 1 min of reaction. Hence, it may indicate that the dimeric structure is not active toward urea hydrolysis.

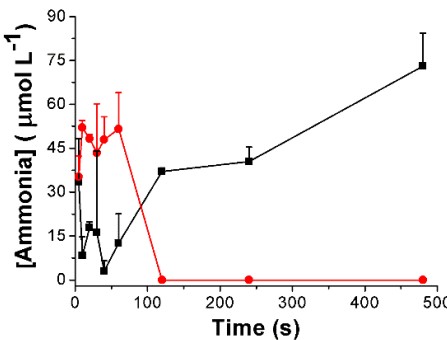

**Figure 10.** Ammonia quantification produced by the [$Cu^{II}L2ClO_4$] complex in the acetonitrile/water (red line) and methanol/water mixture (black line) at 308 K using a 10.4 mmol $L^{-1}$ urea concentration. Only the positive portion of the error bars is shown in the graphic.

In order to verify how the solvent was affecting the equilibrium monomer/dimer, we performed the reaction of urea hydrolysis in other solvents (DMSO, THF, and ethanol), Figures S70–S72. We

observed that a high reaction rate of urea hydrolysis was achieved in solvents that exhibited less pronounced hydrogen bonds with water (ACN, DMSO, and THF) than the organic solvents methanol and ethanol [61]. Therefore, we reasoned that the effect of preferential solvation [62] by the organic solvents in the aquation reaction of our complexes could be tuning the equilibrium dimer/monomer. The interaction between solvent molecules and the complex is probably occurring via the apolar sites of the solvent, since the complex has a neutral charge. This mode of solvation results in a secondary sphere organized in a way that the dipoles of the solvents are oriented to the bulk solution, and water can interact with these sites via hydrogen bonds (Figure 11A). The stronger hydrogen bond between methanol and water stabilizes the initial state of reaction (dimer–solvent), leading to a slower rate of ligand substitution due to the higher activation energy of the reaction in methanol/water (Figure 11B), thus forming less monomers than in ACN (or the DMSO, THF/H$_2$O mixtures). These results are in contrast to the increased reaction rate observed in methanol and ethanol by the other groups [63] due to the stabilization of the transition state, which strengthens our supposition. Moreover, the reaction rate in the ethanol/water mixtures was even slower than in the methanol/water mixtures due to the longer chain of ethanol, resulting in a higher stabilization of the hydrogen bonds with water in the tertiary coordination sphere. Therefore, our analysis of the equilibrium monomer/dimer in solution verified the occurrence of only the monomer species in ACN/H$_2$O by EPR assays, whereas in the MeOH/H$_2$O mixtures, we detected the presence of dimeric structures by HRMS and EPR experiments, corroborating this hypothesis. The thermochemical data obtained from the DFT calculations of the Cu$^{II}$L1 monomer or dimer interacting with a single solvent molecule support that the interaction between the dimeric structure is more stabilized in methanol and water than in acetonitrile. The enthalpic difference between methanol and water was less than 1 kJ/mol, but an eight times higher enthalpy difference was observed between acetonitrile and water (8.5 kJ/mol). Thus, the stabilization of the ground state in strong hydrogen-bond solvents and the competition between solvents is more pronounced in methanol/water systems, which could result in a lower substitution rate of the chloride in methanol/water mixtures in comparison to the acetonitrile/water mixtures, reinforcing our experimental data.

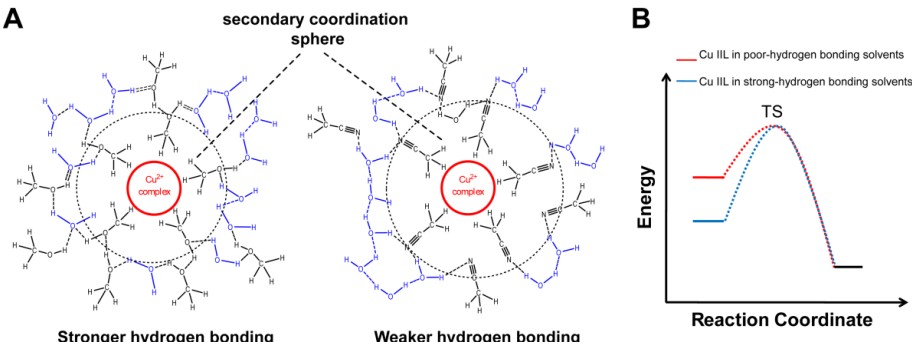

**Figure 11.** Preferential solvation shell of the complexes in methanol/water and acetonitrile/water mixtures (**A**) and its effect in the stabilization of the ground state (dimeric species) (**B**).

Noticing the strong effect of the solvent in the equilibrium monomer/dimer, we suspected that this could enable the tuning of the catalytic behavior by an allosterism (or upregulation).In order to check this possibility, we evaluated complexes Cu$^{II}$L2, Cu$^{II}$L3, and Cu$^{II}$L4 in different urea concentrations (Figure 9 and Figure S68). It should be noted, however, that the increase in urea concentration also increased the water content of the mixture, which could influence the monomer/dimer equilibrium. Noticeably, the complexes presented different behaviors upon an increase in urea (and water) concentrations. For instance: Cu$^{II}$L3 had a sigmoidal behavior, whereas the behavior for Cu$^{II}$L2 was linear (Figure 12). By keeping the water constant at 20%, the water effect in the monomer/dimer equilibrium decreased, and essentially, this effect was more pronounced for Cu$^{II}$L2, observing a 2-fold decrease of the reaction rate when the water concentration was constant. This result indicates that water has a positive

effect in catalysis for the Cu$^{II}$L2 complex due to the shift of inactive dimeric species into active monomeric species. In contrast, the Cu$^{II}$L3 complex and Cu$^{II}$L4, already have their equilibrium shifted to monomeric species and therefore, do not present a strong influence of water in catalysis, even though a slight positive allosteric effect of water is also observed.

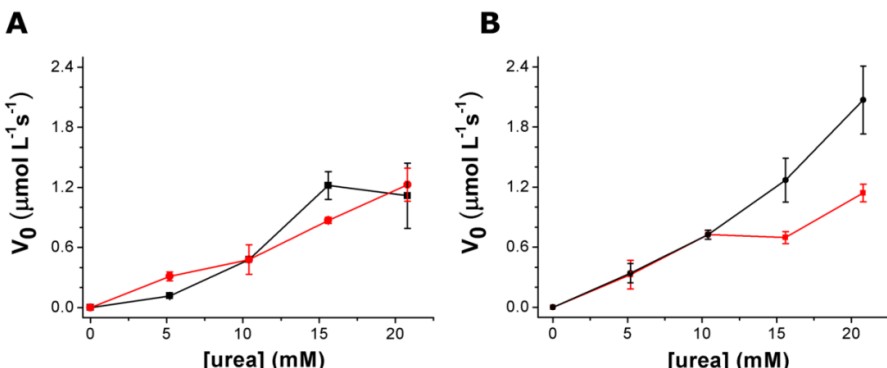

**Figure 12.** Initial rate of urea hydrolysis reaction versus urea concentration performed by Cu$^{II}$L3 (**A**) and Cu$^{II}$L2 (**B**). The black line and squares are relative to the increase in water content of the reaction from 0 to 40%, whereas the red line and circles are relative to the water content kept constant at 20%.

Since other aspects of the reaction could be affecting the reaction kinetics such as the activity of water and reaction pH, we decided to verify their significance in our systems. For instance, the composition of a solvent mixture containing water will change the activity of water ($a_w$) [64] and the increase in $a_w$ could impact the reaction mechanism and kinetics of a hydrolytic reaction. For instance, the water activity of the methanol/water mixtures used in our systems ranged from 0.26 to 0.65 (Figure S76) (determined by the equation from Zhu et al. [65]), whereas the acetonitrile/water (10% volume) mixture had a water activity of 0.9. Therefore, it would be reasonable to ascribe the observed differences of reaction rates to the different water activity of these systems. However, the catalytic reaction performed in ACN containing 2% ($a_w$~0.4) (Figure S74) still showed a high reaction rate of urea hydrolysis in this solvent mixture. This finding reinforces the proposed idea of solvent effect in the reaction kinetics, since acetonitrile aids in the equilibrium shift to monomeric species.

In addition to the water activity, the pH in our reactions was not controlled and the increase in pH during the reaction could affect the coordination sphere and the reaction rate. Taking into consideration this possibility, we performed a reaction including a pH indicator (phenol red) and we did not observe a significant effect in the pH due to the produced ammonia over the reaction time in the UV–Vis spectrum (Figure S73A). To remove any possible interference produced by the indicator, we performed a reaction control without the pH indicator and added it after the reaction reached a saturation level of ammonia. Again, no color change was observed (Figure S73B), indicating that in the methanol/water and acetonitrile/water mixtures (Figure S73C), the produced ammonia did not seem to severely affect the reaction pH. In fact, the pH of the buffered solutions in solvent mixtures has been previously analyzed by other groups, and they noticed that the NH$_3$/NH$_4^+$ buffer presented a lower pH in the solvent mixture than the observed one in pure water [66]. An in situ FTIR experiment (Figure S75) showed an increase of a band in the 3000 cm$^{-1}$ region, indicative of the formation of ammonium ions. After 90 s, we observed that this band oscillated between a minimum and a maximum, suggestive of the equilibrium between ammonia and ammonium. Therefore, due to the fact that the pH of the reaction is not expressively altered during the reaction, we suspect that the main interaction of ammonium is with the hydroxide formed in the hydrolysis reaction of ammonia.

In order to evaluate the impact of the pH in our catalytic reactions, we decided to use buffered aqueous solutions of urea at three different pHs: 3, 6, and 8, and in this case, we observed a profile indicative of a catalyzed reaction at both basic and acidic pHs (Figure S73D). Interestingly, the reaction

rates using buffered solutions were lower than the non-buffered one, which might indicate an inhibition of the hydrolysis by the buffers.

Since the in situ reversibility of the system was not observed, the best classification of the effect of water in the urea hydrolysis reaction is "upregulation" or "regulation" [12]. We believe that the strong water dependence of this system will be able to be explored in the future as water sensors.

## 4. Conclusions

A strong correlation between dimer/monomer equilibrium and catalysis was observed in the copper complexes synthesized in this work. These complexes were synthesized from Schiff bases from L-proline, exhibiting a square planar geometry. EPR analysis enabled us to verify the existence of a mixture of compounds in solution, especially in the methanol/water mixtures. We explored the hydrolytic capacity of these complexes in urea hydrolysis as a model reaction. As observed by EPR, in the acetonitrile/water mixture, the equilibrium shifted to a monomer and the hydrolysis of urea was too fast to detect the kinetics of ammonia formation by the Berthelot method. However, when the equilibrium monomer/dimer was present, as in the methanol/water mixture, the reaction proceeded slower and the ammonia formation kinetics could be detected in a saturation profile. This effect was shown to be due to the preferential solvation effect, by which hydrogen bonds formed between the secondary and tertiary coordination spheres stabilized the initial state of the aquation reaction. A strong influence of water concentration was observed in the methanol/water system, with special attention to $Cu^{II}L2$ complex. A comparison with the dimeric perchlorate complex enabled us to visualize the importance of the monomer in the reaction, since the dimer produced ammonia from urea very slowly. In conclusion, this work relates a key feature of the chlorido bridges in a supramolecular structure of $Cu^{II}$ complexes for an allosteric (upregulation) behavior in catalysis.

**Supplementary Materials:** The following are available online at http://www.mdpi.com/2624-8549/2/2/32/s1 [67–71], Figures S1–S17: NMR spectra of ligands, Figures S18–S22: HRMS of ligands, Figure S23: Optimized crystalline structure and number assignment of HL1, HL2, and HL3, Figures S24–S28: Comparison of FTIR between ligands and complexes, Figure S29: FTIR spectra of $[Cu^{II}L2(CH_3OH)]ClO_4$ dispersed in KBr, Figures S30–S35: Comparison of UV–Vis between ligands and complexes, Figures S36–S53: EPR spectra of complexes, Figures S53–S55: Conductivity measurements of complexes in methanol/water, Figures S56–S62: HRMS of complexes, Figures S63–S68: Ammonia quantification produced by the complexes, Figure S69: Infrared spectra of reactions of the complexes (a) 30 s, (b) 600 s, and with (c) urea. Table S1: Crystal data and structure refinement of L1–L3, Table S2: Bond length for L1–L3, Table S3: Bond angles for L1–L3, Table S4: Comparison of the main infrared bands between ligands and complexes, Table S5: Comparison of transitions in the ultraviolet and visible region between ligands and complexes, Table S6: Comparison of oxidation and reduction potentials due to the cyclic voltammetry of ligands and complexes, Table S7: EPR parameters of aggregates and monomeric species of the $Cu^{II}$ complexes of this work in dichloromethane at 77 K, Table S8: EPR parameters for the $Cu^{II}$ complexes of this work in acetonitrile at 298 K, Table S9: EPR parameters for the $Cu^{II}$ complexes of this work in acetonitrile/water (80/20) mixture at 298 K, Table S10: EPR parameters for the $Cu^{II}L2$ and $[Cu^{II}L2(CH_3OH)]ClO_4$ of this work in the methanol and methanol/water (80/20) mixture at 298 K, Table S11: Maximum amount of ammonia formed by the complexes of this work under the conditions of the acetonitrile/water and methanol/water mixture at 308 K. CCDC 1983221, 1983222 and 1983224 contain the supplementary crystallographic data for L1, L2, and L4, respectively.

**Author Contributions:** Conceptualization, C.B.C. and C.G.C.M.N.; Methodology, C.B.C.; Synthesis, C.B.C.; Characterization, C.B.C., O.R.N., and R.G.S.; Computer simulations, A.F.d.M. and F.M.C.; Formal analysis, C.B.C. and C.G.C.M.N.; Investigation, C.B.C. and C.G.C.M.N.; Resources, C.G.C.M.N., A.F.d.M., and O.R.N.; Data curation, C.B.C.; Writing—original draft preparation, C.B.C., and C.G.C.M.N.; Writing—review and editing, C.B.C. and C.G.C.M.N.; Supervision, C.G.C.M.N.; Project administration, C.G.C.M.N.; Funding acquisition, C.G.C.M.N. and A.F.d.M. All authors have read and agreed to the published version of the manuscript.

**Funding:** This research was funded by FAPESP (projects 2012/15147-4, 2013/07296-2 and 2016/01622-3), CNPq for a master fellowship (132842/2017-3), and CAPES (project 001). Otaciro Rangel Nascimento thanks CNPq (project 305668/2014-5). A.F.d.M. thanks CNPq for a Research Fellowship.

**Acknowledgments:** We would also like to thank Diego E. Sastre for editing the images and Elton Sitta for the aid in the in situ FTIR experiments. The authors acknowledge the National Laboratory for Scientific Computing (LNCC/MCTI, Brazil) and UFSCar for providing the high-performance computing resources of the SDumont supercomputer (http://sdumont.lncc.br) and of the Could@UFSCar, respectively, both of which have contributed to the results reported in this paper.

**Conflicts of Interest:** The authors declare no conflicts of interest.

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
