# Peer review of "Solvent Effect on the Regulation of Urea Hydrolysis Reactions by Copper Complexes"

_chemistry, doi:10.3390/chemistry2020032_

Round 1
Reviewer 1 Report
The manuscript by Marques Netto et al. describes the formation of Cu(II) complexes able to form mono- or dinuclear species following the bridging coordination of a Cl- anion. This latter can be replaced by water or other protic solvents thus disrupting the formation of dinuclear species. The amount of water present in the solvent tunes the stoichiometry of the complexes. The idea is interesting although not unprecedented.
While the synthesis, characterization, and establishment of mono- dinuclear equilibria are sound, the authors fall short in proving that this equilibrium does in fact affect the kinetic of urea hydrolysis. I believe there are a number of reasons why the reported experiments fail to provide convincing and sound evidence of their assumption. Below are my observations.
- The pH of the solution is not controlled. Admittedly, controlling the pH in a mixed solvent could be more challenging than in pure water. Nevertheless, the pH will change following the cleavage process. The change of pH will affect the rate and the coordination sphere of the metal ions.
- Changing the composition of a solvent mixture containing water will change the activity of water as well.
- It is not clear what happens to the released ammonia. Is it fully protonated’ Does it interfere with Cl- in Cu(II) binding? This is a crucial aspect because it will dramatically change the scenario in terms of the species present in the solution.
- The choice of urea as a substrate appears to be quite unfortunate on (at least) two counts: 1) for the reasons outlined in point 3 above; 2) for its very sluggish reactivity. The literature is full of studies using active esters or amides: why don’t they use one of them?
- The authors should become aware of the papers published by R.S. Brown on the role of ethanol and methanol in accelerating metal ion-catalyzed cleavage processes. Prof. Brown retired some time ago, but the authors may refer to one of his latest reviews published in Pure and Applied Chemistry for reference: DOI:10.1515/pac-2014-1008. His observation may affect their analysis.
In conclusion, I cannot clear for publication a paper that claims in its title to show “Solvent effect on the regulation of urea hydrolysis [by controlling the nuclearity of a Cu(II) complex]” when the results reported are not soundly supporting such statement.
Author Response
We would like to kindly thank the revision of the manuscript by the reviewers, since it helped us to improve the quality of our work. In this revision, we addressed each comment of the reviewers and performed more experiments to better indicate the solvent effects in terms of the initial state complex composition towards a model of hydrolytic reaction. The new experiments involved pH control, in situ FTIR, analysis of the influence of the water activity in the reaction and solvent effect (THF, DMSO and ethanol) in the reaction of urea hydrolysis. We verified that the pH of the reaction did not significantly change over the course of the reaction and that ammonium ions are formed, which are probably associated to hydroxide ions, in order to not change the reaction pH. The control of the pH by the use of buffer solutions decreased the reaction rate, indicating a possible inhibition by the buffer ions. The water activity was also addressed in our experiments and we verified that despite of the role action of water in the reaction, the solvent had a greater effect in the catalytic rate. Therefore, we performed experiments in THF, DMSO and ethanol, verifying that our previous results in acetonitrile and methanol are in agreement to the hypothesis of a preferential solvation effect. This effect stabilizes the ground state of the complex in the dimeric form when strong hydrogen bonds are present between water and solvent (methanol or ethanol), slowering the ligand substitution reaction and enabling the visualization of the regulation urea hydrolysis by water.
The manuscript by Marques Netto et al. describes the formation of Cu(II) complexes able to form mono- or dinuclear species following the bridging coordination of a Cl- anion. This latter can be replaced by water or other protic solvents thus disrupting the formation of dinuclear species. The amount of water present in the solvent tunes the stoichiometry of the complexes. The idea is interesting although not unprecedented.
While the synthesis, characterization, and establishment of mono- dinuclear equilibria are sound, the authors fall short in proving that this equilibrium does in fact affect the kinetic of urea hydrolysis.
We have shown that in acetonitrile solution our complexes readily form electrolytes, whereas in methanol solutions a slower rate of ligand substitution was observed. A similar kinetic feature was previously described by Chattopadhyay and Coetzee (doi:10.1021/ic50119a028) in substitution reactions of nickel(II) ion with several ligands (pyridine, 2,2'-bipyridine, 1,1 O-phenanthroline, and 2,2',2l'-terpyridine) using different solvents. In this system, they determined that the substitution of phenanthroline in acetonitrile was much faster than in other solvents, such as methanol, water or DMSO. They reasoned that acetonitrile was intrinsically more disordered than the other solvents and the disruptive effect of solutes was less significant in ACN than in the other solvents. More importantly, this kinetic behavior could be due to a specific interaction between the polarized ACN molecules and phenanthroline occurring in the primary solvation shell of nickel(II), rendering an analogous to the “internal conjugate base mechanism”, thus enhancing the reaction rate. In our case, we verified the effect of preferential solvation (doi:10.1021/ar50115a005) in the aquation reaction of our complexes, in which the high molar fraction of the organic solvent led to a preferential solvation of the complex by the organic molecules instead of water. The interaction between solvent molecules and the complex is probably occurring via the apolar sites of the solvent, since the complex has a neutral charge. This mode of solvation results in a secondary sphere organized in a way that the dipoles of the solvents are oriented to the bulk solution, and water can interact with these sites via hydrogen bonds (figure 9 of the manuscript). The hydrogen bond in methanol/water mixtures is stronger than in acetonitrile/water mixtures (doi:10.1021/acs.jpcb.9b02829), and the stronger hydrogen bond, stabilizes the initial state of reaction (dimer-solvent), leading to a slower rate of ligand substitution and formation of monomers than in the ACN/H2O mixtures, due to the higher activation energy of the reaction in methanol/water. This theory was corroborated by the study of the reaction of urea hydrolysis in other solvents (DMSO, THF and ethanol) in which a high reaction rate (due to high content of monomers) is observed for the poorer hydrogen bonding solvents (DMSO and THF), and a low reaction rate was observed in ethanol. The reaction rate in ethanol/water mixtures was even slower than in methanol/water mixtures, due to the longer chain of ethanol, resulting in a higher stabilization of the hydrogen bonds with water in the tertiary coordination sphere. Therefore, our analysis of the equilibrium monomer/dimer in solution, verified the occurrence of only monomer species in ACN/H2O by EPR assays, whereas in MeOH/H2O mixtures we detected the presence of dimeric structures by HRMS and EPR experiments, corroborating to this hypothesis. In conclusion, we proposed a strong influence of the solvent in the kinetics of urea hydrolysis, due to the significant difference observed between the reactions performed in ACN/H2O and MeOH/H2O mixtures.
These new experiments were included in the support information (Figures S70, S71 and S72) and the discussion was added to the revised manuscript.
I believe there are a number of reasons why the reported experiments fail to provide convincing and sound evidence of their assumption. Below are my observations.
- The pH of the solution is not controlled. Admittedly, controlling the pH in a mixed solvent could be more challenging than in pure water. Nevertheless, the pH will change following the cleavage process. The change of pH will affect the rate and the coordination sphere of the metal ions.
We agree with this reviewer´s observation. Indeed, the pH in our reactions is not controlled and the increase in pH during the reaction could affect the coordination sphere and the reaction rate. Taking into consideration this possibility, we performed a reaction including a pH indicator (phenol red) and we did not observe a significant effect in the pH due to the produced ammonia over the reaction time in the UV-Vis spectrum (Figure S73A). To remove any possible interference produced by the indicator, we performed a reaction control without the pH indicator and added it after the reaction reached a saturation level of ammonia. Again, no color change was observed (Figure S73B), indicating that in the methanol/water and acetonitrile/water mixtures (Figure S73C), the produced ammonia did not seem to severely affect the reaction pH.
In fact, the pH of buffered solutions in solvent mixtures has been analyzed previously by other groups, and they noticed that the NH3/NH4+ buffer presented a lower pH in the solvent mixture than the observed one in pure water (doi: 10.1080/15422110701539129).
In order to evaluate the impact of the pH in our catalytic reactions, we decided to use buffered aqueous solutions of urea at three different pHs: 3, 6 and 8 and in this case we observed a profile indicative of a catalyzed reaction at both, basic and acid pHs (Figure S73D). Interestingly, the reaction rates using buffered solutions were lower than the non-buffered one, which might indicate an inhibition of the hydrolysis by the buffers.
Changing the composition of a solvent mixture containing water will change the activity of water as well.
The reviewer is correct in this statement, since the composition of a solvent mixture containing water will in fact change the activity of water (aw )1. The increase in aw could impact on the reaction mechanism and kinetics of a hydrolytic reaction. For instance, the water activity of the methanol/water mixtures used in our systems ranged from 0.26 to 0.65 (determined by the equation from Zhu et al.2), whereas the acetonitile/water (10% volume) mixture had a water activity of 0.9. Therefore, it would be reasonable to ascribe the observed differences of reaction rates to the different water activity of these systems. However, the catalytic reaction performed in ACN containing 2% (aw~0.4) (Figure S74) still evidences the high reaction rate of urea hydrolysis in this solvent mixture. This finding reinforces the proposed idea of solvent effect in the reaction kinetics, since acetonitrile is aiding in the equilibrium shift to monomeric species.
(1) Blandamer, M. J.; Engberts, J. B. F. N.; Gleeson, P. T.; Reis, J. C. R. Activity of Water in Aqueous Systems; a Frequently Neglected Property. Chem Soc Rev2005, 34 (5), 440-458.
(2) Zhu, H.; Yuen, C.; Grant, D. J. W. Influence of Water Activity in Organic Solvent + Water Mixtures on the Nature of the Crystallizing Drug Phase. 1. Theophylline. Int J Pharm1996, 135 (1-2), 151-160.
- It is not clear what happens to the released ammonia. Is it fully protonated’ Does it interfere with Cl- in Cu(II) binding? This is a crucial aspect because it will dramatically change the scenario in terms of the species present in the solution.
This question is really interesting. In order to comprehend what was happening to the released ammonia, we performed an in situ FTIR experiment (Figure S75). After the addition of urea, it was possible to observe an increase of a band at the 3000 cm-1 region indicative of the formation of ammonium ions. After 90 seconds we observed that this band oscillates between a minimum and a maximum, suggestive of the equilibrium between ammonia and ammonium. We suspect that the main interaction of ammonium is with the hydroxide formed in the hydrolysis reaction of ammonia, since the pH of the reaction was not significantly increased in the reaction, as discussed above.
The choice of urea as a substrate appears to be quite unfortunate on (at least) two counts: 1) for the reasons outlined in point 3 above; 2) for its very sluggish reactivity. The literature is full of studies using active esters or amides: why don’t they use one of them?
As shown above, the pH of the reaction was not affected by the ammonia formation, maintaining it at the acidic region. Moreover, our group is dedicated to the study of urea hydrolysis in functional and structural urease mimetic systems, and in this work we verified the strong influence of the solvent in this reaction, in which the regulation behavior was more pronounced than the urease mimetism. Therefore, only this aspect of the reaction was focused in this manuscript.
The authors should become aware of the papers published by R.S. Brown on the role of ethanol and methanol in accelerating metal ion-catalyzed cleavage processes. Prof. Brown retired some time ago, but the authors may refer to one of his latest reviews published in Pure and Applied Chemistry for reference: DOI:10.1515/pac-2014-1008. His observation may affect their analysis.
We appreciate this comment of the reviewer. However, after evaluation of this paper we arrived to the conclusion that methanol is not accelerating our reactions as observed by Prof. Brown in his review. We must have in mind that the reaction in acetonitrile/water mixtures is much faster than in methanol/water mixtures. Moreover, the reactions in methanol/water have higher rates by the increased content of water, which indicates that a methanolysis is not occurring, due to a key role of water as a reactant (doi:10.1021/ar50115a005). In fact, when no water is present, no reaction is observed, as shown in figure 10B.
In conclusion, I cannot clear for publication a paper that claims in its title to show “Solvent effect on the regulation of urea hydrolysis [by controlling the nuclearity of a Cu(II) complex]” when the results reported are not soundly supporting such statement.
Reviewer 2 Report
The manuscript describes Schiff-base Cu(II) complexes that exist as a mixture of monomeric/dimeric species in different solvents and catalyze urea hydrolysis depending on the monomer/dimer ratio. Although the system seems to have been thoroughly investigated, I don't believe the manuscript has the novelty and significance for publication in Chemistry. In particular, I disagree with the use of the term "allosterism" for the current system. An important feature of biological allosteric enzymes is that the binding of the effector occurs reversibly. In the current system, the catalytic activity is simply regulated by the rate of dimer dissociation, which is an irreversible reaction due to the instability of the chloride-bridged dimer in protic solvents. This catalytic profile bears no functional resemblance to naturally-occurring enzymes and therefore appears rather weak to me as the main attractive point. I therefore do not recommend publication of the manuscript in Chemistry.
Author Response
The manuscript describes Schiff-base Cu(II) complexes that exist as a mixture of monomeric/dimeric species in different solvents and catalyze urea hydrolysis depending on the monomer/dimer ratio. Although the system seems to have been thoroughly investigated, I don't believe the manuscript has the novelty and significance for publication in Chemistry. In particular, I disagree with the use of the term "allosterism" for the current system. An important feature of biological allosteric enzymes is that the binding of the effector occurs reversibly. In the current system, the catalytic activity is simply regulated by the rate of dimer dissociation, which is an irreversible reaction due to the instability of the chloride-bridged dimer in protic solvents.
We apologize for the misunderstanding created by the abstract, which stated that our complexes exhibited an abiotic allosterism. In fact, we did not observe the reversibility of the system and actually, an up regulation in the reaction kinetics was observed. Indeed, we had stated this effect in the title “Solvent effect on the regulation of urea hydrolysis reactions by copper complexes” and the statement of the 405-406 lines “The in-situ reversibility of the system was not observed, therefore the best classification of the effect of water in urea hydrolysis reaction is “up-regulation” or “regulation””.
After performing new experiments, regarding water activity, pH of the reaction and different solvents in the reaction, we reached to the conclusion that a ground state stabilization effect is occurring in our systems due to the preferential solvation shell. From our understanding, this is a new description of the solvent effect on the regulation of reaction kinetics, since it does not involve the increase of electric dipole, but is related to hydrogen bond forming in the ground state (see discussion and figure 9 in the manuscript).
This catalytic profile bears no functional resemblance to naturally-occurring enzymes and therefore appears rather weak to me as the main attractive point. I therefore do not recommend publication of the manuscript in Chemistry.
It is unclear to us what the reviewer means with this statement: if it is about the catalysis reaction (urea hydrolysis) or the regulation of a catalytic reaction. If the reviewer meant the hydrolysis reaction, it should be noticed that we were not trying to mimic a specific enzymatic reaction. On the contrary, we explored the regulation behavior of a complex by solvation effects, which is in fact, related to regulatory proteins, since solvation of a protein can change its tertiary structure and it is known that changes in the tertiary structure of some proteins can impact in the kinetics of exchange reactions, such as in hemoglobin (doi:10.1073/pnas.93.25.14526 and doi:10.1073/pnas.93.25.14526), an allosteric protein. Therefore, in our system the resemblance to a natural occurring enzyme is based on the effects of solvation on the kinetic profile of the catalytic reaction.Moreover, the dimer/monomer equilibrium is also seen in naturally occurring systems, as a regulatory function, such as in thymidylate synthase (doi:10.1002/pro.379 10.3390/ijms19051393)and other proteins.
Reviewer 3 Report
This manuscript contains solvent effect on the regulation of urea hydrolysis reactions by the use of dimeric Schiff-base copper complexes connected by a chlorido ligand bridge.
The background as well as the results are well presented and the authors take great care to limit a key feature of the chlorido bridges in a supramolecular structure of CuII complexes for an allosteric (up-regulation) behavior in catalysis. Thus, this manuscript could be interesting for the readers in the field of supramolecular chemistry as well as organometallic catalysts. Also this paper could well serve as the basis for a development of these fields. The authors improved the original manuscript by following the referee’s comments.
Minor comments
- In Figure 9 the authors proposed the preferential salvation shell of the complexes in methanol/water and acetonitrile/water mixtures (A) and its effect in the stabilization of the ground state (dimeric species) (B). They should explore the supporting evidence about the stabilization of the ground state arising from the intermolecular hydrogen bonding with solvents in the system by using, such as computational studies to evaluate the energy levels of the ground state.
- The melting points of the all synthesized compounds are required.
- The authors should improve the typos (space problems, many), hyphenation, and capital letters in the text.
I recommend this manuscript for publication in Chemistryafter minor revisionsdescribed above.
Author Response
This manuscript contains solvent effect on the regulation of urea hydrolysis reactions by the use of dimeric Schiff-base copper complexes connected by a chlorido ligand bridge.
The background as well as the results are well presented and the authors take great care to limit a key feature of the chlorido bridges in a supramolecular structure of CuII complexes for an allosteric (up-regulation) behavior in catalysis. Thus, this manuscript could be interesting for the readers in the field of supramolecular chemistry as well as organometallic catalysts. Also this paper could well serve as the basis for a development of these fields. The authors improved the original manuscript by following the referee’s comments.
Minor comments
- In Figure 9 the authors proposed the preferential salvation shell of the complexes in methanol/water and acetonitrile/water mixtures (A) and its effect in the stabilization of the ground state (dimeric species) (B). They should explore the supporting evidence about the stabilization of the ground state arising from the intermolecular hydrogen bonding with solvents in the system by using, such as computational studies to evaluate the energy levels of the ground state.
We have contacted a collaborator to perform computational studies regarding the energy levels of the ground state. All the performed analyses were added in the support information and the most important features of this study were inserted in the main text of the manuscript. In essence, due to time constraints, it is impossible to perform a computational study regarding the influence of the solvent mixture, but we were able to obtain thermochemical data from the DFT calculations of the CuIIL1 monomer or dimer interacting with a single solvent molecule. These experiments support that the interaction between the dimeric structure is more stabilized in methanol and water rather than in acetonitrile. The enthalpic difference between methanol and water is less than 1 kJ/mol, but an eight times higher enthalpy difference was observed between acetonitrile and water (8.5 kJ/mol), Table S12. Thus, the competition between solvents is more pronounced in methanol/water systems, which could result in a lower substitution rate of the chloride in methanol/water mixtures in comparison to the acetonitrile/water mixtures, reinforcing our experimental data.
- The melting points of the all synthesized compounds are required.
We would gladly perform the analysis of the melting point of our compounds, however, due to the COVID-19 pandemia the university is closed for non related research to this disease. We expect the reviewer to comprehend this situation.
- The authors should improve the typos (space problems, many), hyphenation, and capital letters in the text.
We properly corrected this issue.
Round 2
Reviewer 1 Report
I appreciate the work done by the authors to properly address the questions I had raised in my previous review. Indeed, some of my points have been now clarified. There are, regrettably, a few open questions that deserve careful attention in a further revision of the manuscript.
As acknowledged by the authors there are several parameters affecting the reaction kinetics. The authors support one of them, but they are still not clear enough in analyzing the others to rule their role out.
Incidentally, there is no trace in the Experimental section of how the kinetics have been performed. Only ammonia quantification is mentioned. The statement (lines 412-414) “Moreover, the reaction in methanol/water only started to produce ammonia after 5 minutes of reaction, whereas CuIIL2 complex was able to produce it after 1 minute of reaction. Hence, it may indicate that the dimeric structure is not active towards urea hydrolysis” may suggest that the reaction is started by adding water. But why don’t they settle the problem by first adding water, let the complex equilibrate, and only at the end add urea? Does this way of proceeding remove the lag time?
The major parameters that need to be carefully evaluated are: pH and water activity. In addition, mixed solvents may preferentially solvate protonated/neutral species.
1. The question of pH is not settled. Although the authors do not suggest a catalytic mechanism for the hydrolysis reaction it is obvious that they operate under acid catalysis. The starting (unbuffered) solution should be acidic (how much?) and hence the addition of phenol red as an indicator (pKa 8 in water) is probably not the best choice. Suppose the original pH was 4 and it shifts to 6. That would not be detected by the indicator. In the case of acid catalysis the reaction could slow down up to two orders of magnitude during the hydrolysis.
2. I don’t understand why there is not a graph reporting the rate (determined as they prefer: initial rate, absolute concentration at a given time…) of ammonia evolution vs the solvent composition (I understand it is not feasible in acetonitrile and the other non-protic solvents, but it should be feasible for the alcohols). This would allow one to appreciate the role of water. I would like to see also the number corrected for the actual concentration of water which is involved in the reaction. This would make life easier for the reader and more clear the message. (How much does the rate change? The authors themselves contributed to the confusion by writing in the abstract: “It was evident that the dimeric species were inactive and that by increasing the water concentration in the reaction medium, the dimeric structures dissociated to form the active monomeric structures. This behavior was more pronounced when methanol/water mixtures were employed, due to a slower displacement of the chlorido bridge in this medium than in acetontitrile/water mixtures, enabling the reaction kinetics to be evaluated” and later (lines 422-424) “We observed that a high reaction rate of urea hydrolysis was achieved in solvents that exhibited less pronounced hydrogen bonds with water (ACN, DMSO and THF), than the organic solvents methanol and ethanol” The two statements appear in contradiction. The point is that the abstract stresses NOT the kinetics but the rate of the switch of the equilibrium.
3. The fact that upon addition of a less solvating solvent favors ammonia vs ammonium is obvious. This would imply the more water there is the less ammonia is present in solution. Ammonia does compete with water in the coordination to Cu(II). This likely affects the reactivity, too.
Finally, what does the following statement mean (lines 482-486): “After 90 seconds we observed that this band oscillates between a minimum and a maximum, suggestive of the equilibrium between ammonia and ammonium. Therefore, owing to the fact that the pH of the reaction is not expressively altered during the reaction, we suspect that the main interaction of ammonium is with the hydroxide formed in the hydrolysis reaction of ammonia.” Are the authors suggesting the protonation/deprotonation of species is so slow to be followed by IR? Why is this an oscillating reaction?
I look forward to receiving the properly revised manuscript.
Author Response
I appreciate the work done by the authors to properly address the questions I had raised in my previous review. Indeed, some of my points have been now clarified. There are, regrettably, a few open questions that deserve careful attention in a further revision of the manuscript.
As acknowledged by the authors there are several parameters affecting the reaction kinetics. The authors support one of them, but they are still not clear enough in analyzing the others to rule their role out.
Incidentally, there is no trace in the Experimental section of how the kinetics have been performed. Only ammonia quantification is mentioned.
Indeed, the reviewer is again correct in the statements. We fixed this problem, by adding the exact procedure used in the kinetics and ammonia quantification, please see lines 219-243.
The statement (lines 412-414) “Moreover, the reaction in methanol/water only started to produce ammonia after 5 minutes of reaction, whereas CuIIL2 complex was able to produce it after 1 minute of reaction. Hence, it may indicate that the dimeric structure is not active towards urea hydrolysis” may suggest that the reaction is started by adding water. But why don’t they settle the problem by first adding water, let the complex equilibrate, and only at the end add urea? Does this way of proceeding remove the lag time?
We do not understand the idea of removing the lag time, since the lag time was indicating that the dimeric specie were inactive towards urea hydrolysis, corroborating to our hypothesis. The reaction of urea hydrolysis is enabled only when water displaces the chloride bridge, forming the aquo-monomeric specie, which is the active compound towards urea hydrolysis. Therefore, if we first equilibrate the system with water, monomeric species would be formed, enabling a fast kinetics, which would not give us the information regarding the activity of the dimer.
The major parameters that need to be carefully evaluated are: pH and water activity. In addition, mixed solvents may preferentially solvate protonated/neutral species.
- The question of pH is not settled. Although the authors do not suggest a catalytic mechanism for the hydrolysis reaction it is obvious that they operate under acid catalysis. The starting (unbuffered) solution should be acidic (how much?) and hence the addition of phenol red as an indicator (pKa 8 in water) is probably not the best choice. Suppose the original pH was 4 and it shifts to 6. That would not be detected by the indicator. In the case of acid catalysis the reaction could slow down up to two orders of magnitude during the hydrolysis.
The reaction operates both under basic and acid conditions, as seen by the V-profile in Figure S73. The employed unbuffered solution had a pH of 6 (this information was added in line 220). Also, phenol red has a pH range of 6 to 8.2 and considering we are starting at pH 6, we would be able to observe in the UV-Vis spectrum if any significant pH change were occurring in the reaction. We agree that the kinetics could be affected during catalysis by pH change, but we do not see this influencing our conclusions on the solvent effect in the reaction. Moreover, the buffered solution at pH 6 (rate of 0.3 molL-1 s-1) had a four times slower kinetics than the unbuffered one (rate of 1.2 molL-1 s-1) , which indicates a higher impact of the buffer on kinetics rather than the increasing pH over the reaction time.
- I don’t understand why there is not a graph reporting the rate (determined as they prefer: initial rate, absolute concentration at a given time…) of ammonia evolution vs the solvent composition (I understand it is not feasible in acetonitrile and the other non-protic solvents, but it should be feasible for the alcohols). This would allow one to appreciate the role of water. I would like to see also the number corrected for the actual concentration of water which is involved in the reaction. This would make life easier for the reader and more clear the message. (How much does the rate change?
We added a graphic in the support information (Figure S76) in which the initial rate is plotted versus water activity for both CuL2 and CuL3 complexes. For the ethanol/water mixture composition there is only one concentration of water (20%), therefore, we do not think that a graphic would be valuable here. The initial rate of the ethanol/water mixture was of 0.45 molL-1 s-1, whereas for methanol/water mixture, the rate is doubled, as readily seen in the kinetic profiles of figures S71 and figure 7B.
The authors themselves contributed to the confusion by writing in the abstract: “It was evident that the dimeric species were inactive and that by increasing the water concentration in the reaction medium, the dimeric structures dissociated to form the active monomeric structures. This behavior was more pronounced when methanol/water mixtures were employed, due to a slower displacement of the chlorido bridge in this medium than in acetontitrile/water mixtures, enabling the reaction kinetics to be evaluated” and later (lines 422-424) “We observed that a high reaction rate of urea hydrolysis was achieved in solvents that exhibited less pronounced hydrogen bonds with water (ACN, DMSO and THF), than the organic solvents methanol and ethanol” The two statements appear in contradiction. The point is that the abstract stresses NOT the kinetics but the rate of the switch of the equilibrium.
Urea hydrolysis kinetics is directly connected to the switch of the equilibrium. If the equilibrium is directed to the monomer formation, as it is in less-pronounced hydrogen bond solvents (ACN, DMSO and THF), the reaction rate is higher.
- The fact that upon addition of a less solvating solvent favors ammonia vs ammonium is obvious. This would imply the more water there is the less ammonia is present in solution. Ammonia does compete with water in the coordination to Cu(II). This likely affects the reactivity, too.
Assuming the reviewer is concerned with the less hydrogen bonded solvents, we can inspect the results for acetonitrile/water mixtures (Figures 7A and S74). In figure S74 we have the least water content, and according to the reviewer, would indicate higher ammonia content, over ammonium. The reviewer suggests that ammonia might compete with water and therefore, the reactivity over urea hydrolysis would be affected (negatively). However, we still observed a high (and fast) reactivity in figure S74, reaching the same quantity of ammonia (around 250 µmolL-1) than the one observed in figure 7A (blue line). Moreover, we did not observe ammonia coordination in any of our experiments by FTIR, therefore, we believe that the ammonia formation is not affecting the reactivity.
Finally, what does the following statement mean (lines 482-486): “After 90 seconds we observed that this band oscillates between a minimum and a maximum, suggestive of the equilibrium between ammonia and ammonium. Therefore, owing to the fact that the pH of the reaction is not expressively altered during the reaction, we suspect that the main interaction of ammonium is with the hydroxide formed in the hydrolysis reaction of ammonia.” Are the authors suggesting the protonation/deprotonation of species is so slow to be followed by IR? Why is this an oscillating reaction?
In this reaction there is two equilibrium reactions: OH-(aq) + NH4+ (aq) D NH3(aq) + H2O(l) and NH3 (aq) D NH3(g). Once gaseous ammonia if released from the system, more ammonia has to be formed in order to restore the equilibrium, and this could be one of the reasons behind the oscillation. Another reason would be that the second reaction product, possibly isocyanate, can react with ammonia, reducing temporarily its concentration. However, we would need to perform several experiments to better comprehend these oscillations, which is not the scope of this work. We assure that we will try to address this question in a future work.
I look forward to receiving the properly revised manuscript.
Round 3
Reviewer 1 Report
In reply to my comment: "But why don’t they settle the problem by first adding water, let the complex equilibrate, and only at the end add urea? Does this way of proceeding remove the lag time?" the authors state: "[...] Therefore, if we first equilibrate the system with water, monomeric species would be formed, enabling a fast kinetics, which would not give us the information regarding the activity of the dimer." In my opinion that experiment would strongly support the evidence that the break down of the dimer does in fact (experimentally!) affect positively the kinetics. This would be a direct (missing) further evidence that the dimer/monomer equilibrium affects the kinetics. Chemistry relies on direct evidences.
Author Response
In reply to my comment: "But why don’t they settle the problem by first adding water, let the complex equilibrate, and only at the end add urea? Does this way of proceeding remove the lag time?" the authors state: "[...] Therefore, if we first equilibrate the system with water, monomeric species would be formed, enabling a fast kinetics, which would not give us the information regarding the activity of the dimer." In my opinion that experiment would strongly support the evidence that the break down of the dimer does in fact (experimentally!) affect positively the kinetics. This would be a direct (missing) further evidence that the dimer/monomer equilibrium affects the kinetics. Chemistry relies on direct evidences.
The reviewer is correct to state that chemistry relies on direct evidence and for that, the reviewer proposed one experiment to be performed to give direct evidence that the dimer/monomer equilibrium affect the kinetics. Actually, we would like to point that we have direct evidence that the dimer/monomer equilibrium affect the kinetics. For instance, when we characterized the complex CuIIL2 by HRMS both in DCM/ACN (Figure S57) and in DCM/methanol mixtures (Figure S58), we observed,in the mixture containing acetonitrile, an ESI+ spectrum containing two main peaks, one at m/z 508.1415 (calculated for [M-HCl-HOMe]+ 508.1576) and the other at m/z 542.1031 (calculated for [M-HOMe]+ 542.1181), ascribed for the monomer. In contrast, the HRMS ESI+ spectrum in CH2Cl2/CH3OH had a peak at m/z 1169.2410, which is ascribed for the dimer. Interestingly, in these experiments, only a limited amount of water was present, but in the DCM/ACN mixture, it was sufficient to generate only monomers, corroborating with our DFT thermochemical data (Table S12). Hence, in ACN/water mixtures, only monomers are present in solution, as evidenced by EPR and conductivity measurements (Figures S46, 2) , resulting in fast kinetics, observed in figures 8, 9, S63,S64,S65 and S66. In addition, in the DFT experiments it was observed that the dimeric structure is more stabilized in methanol and water solvents rather than in acetonitrile (Table S12), as also evidenced by EPR, conductivity and HRMS(Figures 6 ,S53 andS58). Therefore, the competition between solvents is more pronounced in methanol/water systems, which can result in a lower substitution rate of the labile ligand in methanol/water mixtures in comparison to the acetonitrile/water mixtures. Altogether, these evidences support the conclusion that dimer/monomer equilibrium affects the kinetics.